# ADAMTS2 promotes radial migration by activating TGF-β signaling in the developing neocortex

Noe Kaneko[1,2], Kumiko Hirai[1], Minori Oshima[1,2], Kei Yura [ID] [2,3], Mitsuharu Hattori [ID] [4], Nobuaki Maeda[1] & Chiaki Ohtaka-Maruyama [ID] [1✉]

## Abstract

The mammalian neocortex is formed by sequential radial migration of newborn excitatory neurons. Migrating neurons undergo a multipolar-to-bipolar transition at the subplate (SP) layer, where extracellular matrix (ECM) components are abundantly expressed. Here, we investigate the role of the ECM at the SP layer. We show that TGF-β signaling-related ECM proteins, and their downstream effector, p-smad2/3, are selectively expressed in the SP layer. We also find that migrating neurons express a disintegrin and metalloproteinase with thrombospondin motif 2 (ADAMTS2), an ECM metalloproteinase, just below the SP layer. Knockdown and knockout of Adamts2 suppresses the multipolar-to-bipolar transition of migrating neurons and disturbs radial migration. Time-lapse luminescence imaging of TGF-β signaling indicates that ADAMTS2 activates this signaling pathway in migrating neurons during the multipolar-to-bipolar transition at the SP layer. Overexpression of TGF-β2 in migrating neurons partially rescues migration defects in ADAMTS2 knockout mice. Our data suggest that ADAMTS2 secreted by the migrating multipolar neurons activates TGF-β signaling by ECM remodeling of the SP layer, which might drive the multipolar to bipolar transition.

Keywords Cerebral Cortex; ADAMTS2; Radial Migration; TGF-β; ECM
Subject Categories Development; Neuroscience

## Introduction

The mammalian neocortices are densely populated with neurons of different characteristics and morphologies within their six layers, which are responsible for cognition, memory, and motor behaviors (Rakic, 1974). Defects in the cortical layer formation processes lead to various brain malformations, and neurological and psychiatric disorders in humans (Gleeson and Walsh, 2000; Stolp et al, 2012). During fetal development of the neocortex, excitatory neurons are born in the ventricular zone and travel long distances toward the pial surface, during which two migration modes, multipolar migration and locomotion, are used sequentially (Nadarajah et al, 2001; Tabata and Nakajima, 2003) (Fig. 1A). This change of migration mode accompanies the multipolar-to-bipolar transition of migrating neurons at the subplate (SP) layer. Recently, we revealed that SP neurons (SpN), first-born, and matured neurons in the developing neocortex, form transient synapses on subsequently born multipolar neurons (MpN) (Ohtaka-Maruyama et al, 2018). The glutamatergic synaptic transmission from SpNs to MpNs induces calcium signaling in the MpNs, leading to their multipolar-to-bipolar transition and change of migration mode. Although many genes have been reported to control the radial neuronal migration process (Ohtaka-Maruyama and Okado, 2015), little is known about the molecular mechanism that induces the multipolar-to-bipolar transition after this synaptic transmission.

The SpNs reside in the SP layer, which contains abundant extracellular matrix (ECM) components, including chondroitin sulfate proteoglycans (CSPG) such as neurocan, phosphacan, and versican (Maeda, 2015; Popp et al, 2003). It is known that the ECM provides mechanical support to the tissues, and plays important roles in cell differentiation, adhesion, migration, and cell-cell interaction. In addition to the multidomain large glycoproteins, the ECM contains a variety of growth factors and cytokines, such as FGF and TGF-β, and regulates the stability and bioactivities of these signaling molecules (Kim et al, 2011). TGF-β is sequestered in the ECM as an inactive dimer bound to the latent TGF-β-binding protein (LTBP) by a latency-associated peptide (LAP), which is called a large latent complex (LLC). LLCs also bind to the ECM components, such as fibronectin fibers and fibrillin microfibrils, forming very large complexes (Lockhart-Cairns et al, 2022; Rifkin et al, 2022). Activation of latent TGF-β is tightly regulated by many factors, such as integrins (Lockhart-Cairns et al, 2022) and ECM remodeling induced by proteolytic processing of LLCs (Tatti et al, 2008), which leads to the release of active TGF-β. The released disulfide-bonded TGF-β dimer binds to two TGF-βRII receptors, which recruits two TGFβ-RI receptors, and this heterotetrameric receptor complex initiates the downstream Smad-dependent signaling (Huang and Chen, 2012).

Many studies have linked the CSPGs in the SP layer with axon guidance (Han et al, 2019; Maeda, 2015; Soleman et al, 2013), and our previous studies indicated that CSPGs play important roles in

[1]Developmental Neuroscience Project, Department of Brain & Neurosciences, Tokyo Metropolitan Institute of Medical Science, Tokyo, Japan. [2]Department of Life Science, Graduate School of Humanities and Sciences, Ochanomizu University, Tokyo, Japan. [3]School of Advanced Science and Engineering, Waseda University, Tokyo, Japan. [4]Department of Biomedical Science, Graduate School of Pharmaceutical Sciences, Nagoya City University, Nagoya, Japan. ✉E-mail: maruyama-ck@igakuken.or.jp

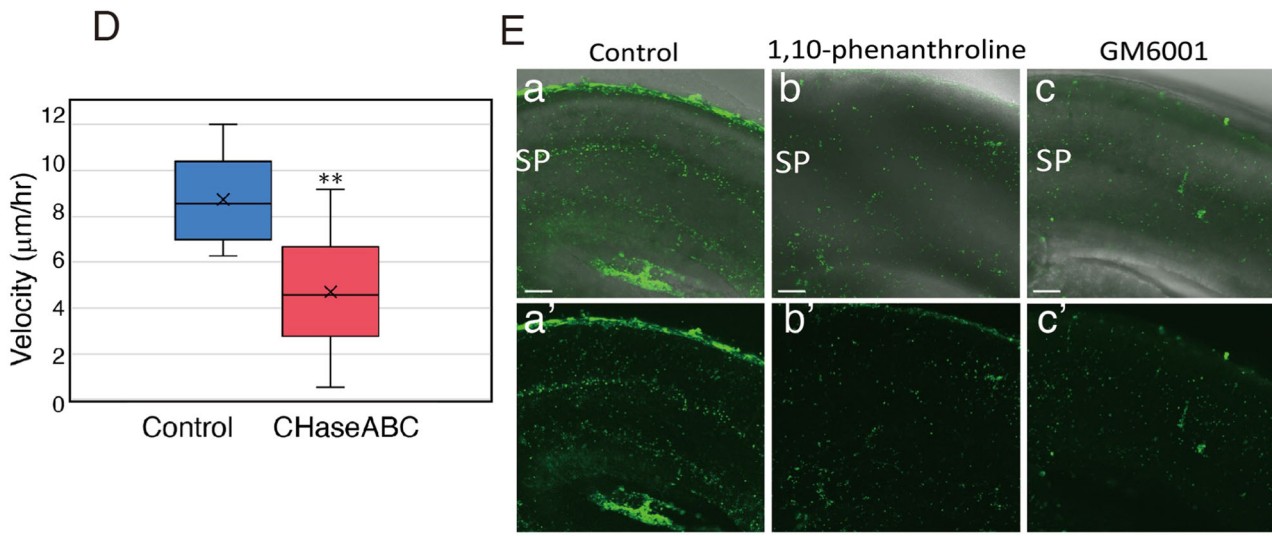

**Figure 1. Radial neuronal migration and the ECM molecules expressed in the SP layer.**

(A) A schematic model of radial neuronal migration in the neocortex (left) with a schematic representation of the distribution of various ECM molecules around the SP layer (right). (B) Immunostaining with antibodies for neurocan confirmed that neurocan is specifically highly expressed in the SP layer. RT-PCR using E16 cortical RNA revealed that the versican V0 isoform is a primary subtype. The immunohistochemical signals of cleaved versican were also detected in the SP layer (arrowheads). The upper dashed line shows the cerebral surface, and the lower dashed line shows the border of the ventricular surface. (C) CHase ABC treatment disturbed radial neuronal migration. Slices prepared from E16.5 cortices (EP of GFP at E14.5) were cultured with or without CHaseABC for 3 days (left). In the control slices, many GFP-labeled neurons migrated to the upper part of the cortex, whereas neurons were accumulated beneath the SP layer in the CHaseABC-treated slices. Time-lapse imaging of control migrating neurons revealed that neurites that did not become leading process degenerated (white arrows), whereas neurites became thicker if determined to be leading process (yellow arrows). In contrast, CHase ABC-treated neurons remained multipolar without leading processes. Time-lapse imaging was performed for 50 h after adding CHaseABC; images for 3 h after 32.5 h are shown. (D) Vertical migration velocity of neurons in the CHaseABC-treated slices. Migration velocity was measured by tracking from time-lapse imaging, and ten cells per group were measured. (E) In situ zymography using DQ-gelatin revealed ECM protease activities in the SP layer. FITC signals were visualized by proteolysis of DQ-gelatin. The MZ and SP layer were distinctly FITC-positive (a, a'). b–c', The slices incubated with metalloproteinase inhibitors (1,10-phenanthroline and GM6001) did not show these signals (b–c'). Data information: In (D), Each group plots measurement data from 10 cells each. The statistical significance was measured by unpaired, two-tailed t-tests (*$p < 0.05$; **$p < 0.01$; ***$p < 0.001$). Scale bars: (B) 50 μm; (C-left and E) 100 μm; (C-right) 10 μm; MZ marginal zone, CP cortical plate, SP subplate, IZ intermediate zone, VZ ventricular zone.

early neuronal polarization and migration (Ishii and Maeda, 2008; Nishimura et al, 2010). However, the roles of the SP layer ECM in neuronal migration remain unclarified. As mentioned above, the SP layer is a strategically important region where the multipolar-to-bipolar transition occurs during radial neuronal migration. Thus, we investigated how the ECM of the SP layer is involved in the morphological changes of migrating neurons. In this study, we found that the expression of ADAMTS2 (a disintegrin and metalloproteinase with thrombospondin motif 2) in the migrating neurons was significantly upregulated just below the SP layer during the multipolar-to-bipolar transition and is required for the radial neuronal migration. Although ADAMTS2 is a well-known procollagen-cleaving enzyme and a causative gene for Ehlers-Danlos syndrome (EDS), a skin disorder caused by abnormal collagen fibrillogenesis (Colige et al, 1999; Jaffey et al, 2019), Bekhouche et al recently revealed that TGF-β signaling is one of the targets of ADAMTS2 (Bekhouche et al, 2016). The substrates of ADAMTS2,3 included LTBP-1, TGF-βRIII, and ADAMTS2 activated TGF-β signaling in the human fibroblasts. Therefore, we focused on TGF-β signaling as a downstream candidate of ADAMTS2. We found that TGF-β signaling is activated in the migrating neurons just below the SP layer, in which ADAMTS2 is essential for the activation of the signaling. We suggest that the ECM remodeling induced by ADAMTS2 secreted by migrating neurons activates TGF-β signaling at the SP layer, which leads to the multipolar-to-bipolar transition and the change of the migration mode (Fig. 7H).

## Results

### ECM molecules and remodeling in the SP layer

It has been reported that CSPGs, such as neurocan and versican, localize in the SP layer of the developing cortex (Oohira et al, 1994; Popp et al, 2003). We used existing databases to search for the other ECM molecules selectively expressed in the SP layer of the developing neocortex. In situ hybridization data (Allen Brain Atlas and Eurexpress) revealed that phosphacan/PTPζ (PTPRZ1) is highly expressed at the SP layer (Fig. EV1A). In addition, fibronectin 1, collagen XIα1, LTBP 1, and fibrillin 2 are also expressed at the SP layer (Fig. EV1A,B). It is also known that there are multiple versican isoforms (Ito et al, 1995; Lemire et al, 1999).

We investigated which type is produced in the developing cortex by RT-PCR and confirmed that full-length V0 is the predominant isoform (Fig. 1B, right). We then performed immunostaining of CSPGs in the neocortical sections. The results showed that besides neurocan, the cleaved versican (McCulloch et al, 2009) localized at the SP layer (Fig. 1B), suggesting that cleavage of versican, presumably the V0 isoform, occurs around SpNs. The ECM is not a static structure but is dynamically remodeled during various biological processes, especially during development (Koledova et al, 2016; Neupane et al, 2022; Schedin and Keely, 2011). To examine whether ECM remodeling is involved in radial neuronal migration, we cultured cortical slices of mice from embryonic day 16 (E16), in which neurons were electroporated with GFP-expression plasmids at E14, for 3 days in the presence of chondroitinase ABC (CHase ABC). CHase ABC is a bacterial enzyme that degrades chondroitin sulfate and hyaluronan, and thus, breaks down ECM structures (Appendix Fig. S1) and may affect the ECM-dependent signaling processes. The results showed that there was a significant delay in neuronal migration when CHase ABC was added (Fig. 1C, left panels). Time-lapse imaging showed that the multipolar-to-bipolar transition hardly occurred, and the leading processes were not easily determined in the treated neurons (Fig. 1C, right panels, Movie EV1). When we measured vertical migration velocity toward the brain surface using time-lapse imaging data, the velocity of migrating cells in the CHaseABC-treated slices was slower than that in the control slices (Fig. 1D). These results suggested that the normally regulated ECM plays an essential role in the multipolar-to-bipolar transition and radial neuronal migration.

Next, we tried to visualize the ECM proteolytic activity in the developing cortex by using in situ zymography with DQ-gelatin. DQ-gelatin is a fluorogenic substrate consisting of highly quenched, fluorescein-labeled gelatin (Mook et al, 2003). Upon proteolytic degradation, green fluorescence is revealed, and thus, can be used to localize proteolytic activities in the ECM. As a result, we found that the migrating neurons exhibited gelatinolytic activities in the SP layer, which were suppressed by the metalloproteinase inhibitors, 1,10-phenanthroline and GM6001 (Fig. 1E). When migrating neurons were labeled with RFP by in utero electroporation at E14.5 and time-lapse imaging of their gelatinase activity was examined in detail, migrating neurons that entered the SP layer showed a green fluorescent signal, whereas cells before entering or after exiting the SP layer showed no signal (Fig. EV2A,B). These results show that migrating neurons show increased gelatinase activity, specifically in the SP layer.

## A

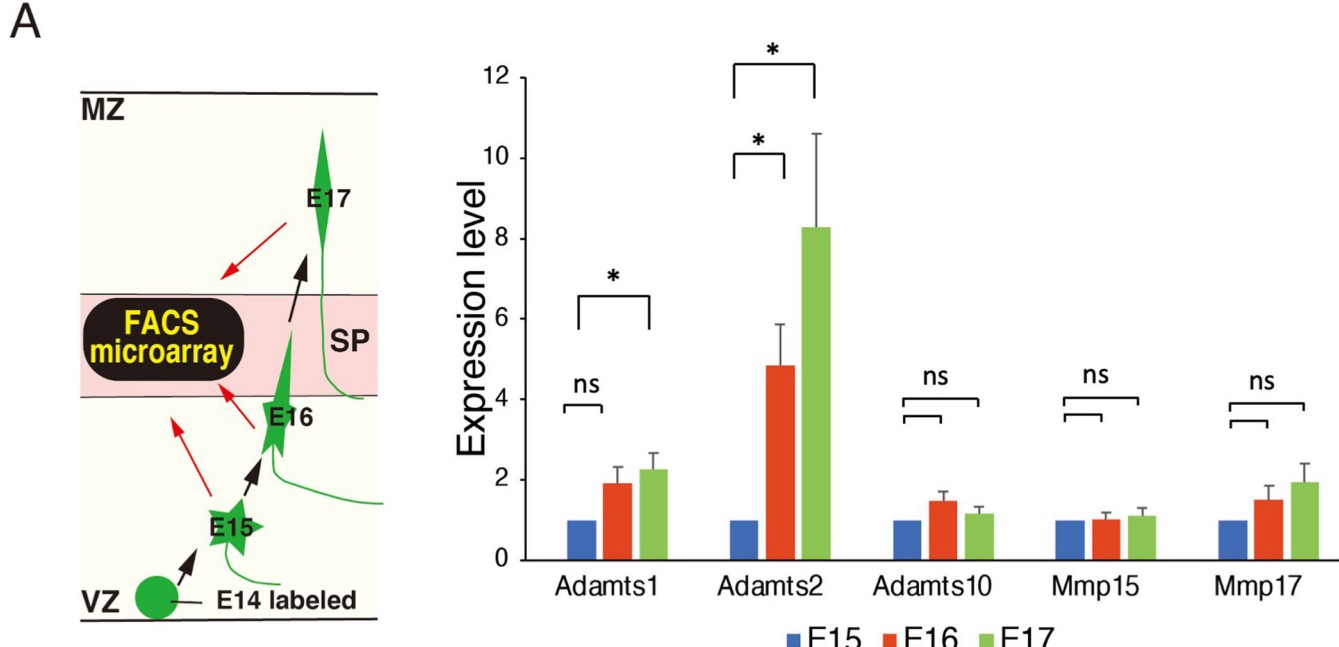

## B

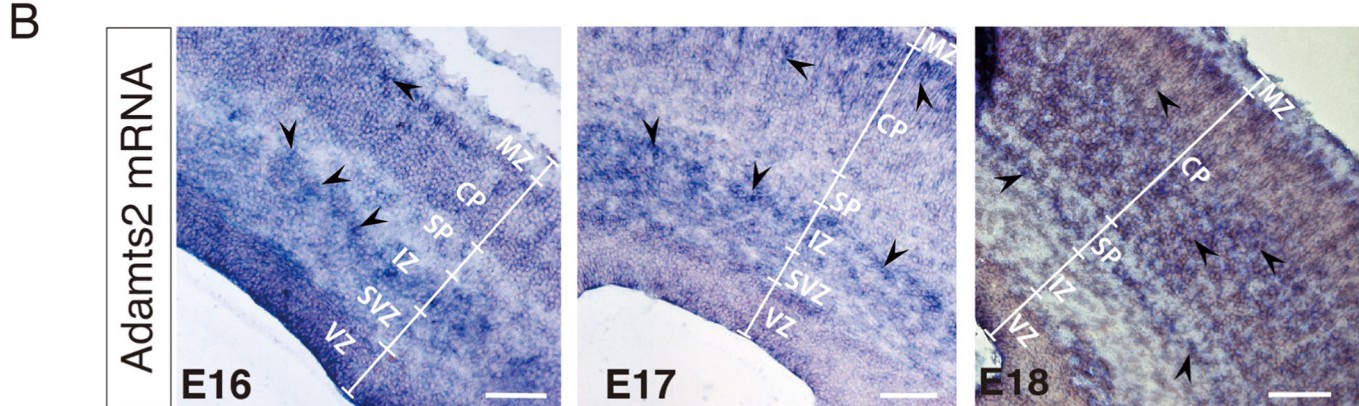

## C

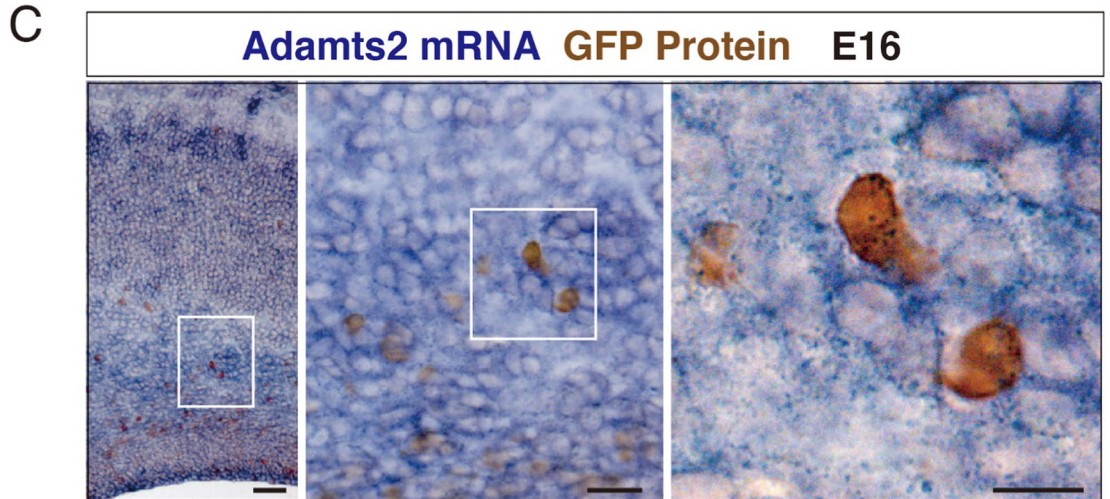

Figure 2. Migrating neurons express ADAMTS2 in the SP layer.

(A) Left: A schematic representation of the FACS microarray analyses of migrating neurons. Right: FACS microarray analyses revealed that the expression of Adamts2 mRNA was increased during radial migration. (B) Adamts2 mRNA was localized at the upper part of IZ, the lower part of the SP layer, and the CP. Arrowheads refer to mRNA signals. (C) Double staining of ADAMTS2-ISH and anti-GFP antibody-IMH revealed that GFP-positive migrating neurons express Adamts2 mRNA. Data information: In (B), data represents the average values from three independent experiments. The statistical significance for each pair was measured by unpaired, two-tailed t-tests (*$p < 0.05$; **$p < 0.01$; ***$p < 0.001$). Scale bars: (B) 100 µm; (C-left) 50 µm; (C-middle) 20 µm; (C-right) 10 µm. MZ marginal zone, CP cortical plate, SP subplate, IZ intermediate zone, SVZ subventricular zone, VZ ventricular zone.

These results indicated that the ECM components and proteolytic activities are present in the SP layer, supporting the hypothesis that the ECM remodeling contributes to the multipolar-to-bipolar transition during radial neuronal migration.

## Adamts2 is expressed in radially migrating neurons

Next, we used microarray data to investigate which genes are involved in the ECM remodeling. To date, we have performed microarray analyses to reveal the changes in gene expression of the migrating neurons during radial migration (Ohtaka-Maruyama et al, 2018). We introduced GFP-expression plasmid into the neural progenitor cells at E14 by in utero electroporation, and then the GFP-labeled neurons were separated by fluorescence-activated cell sorting (FACS) analysis at E15, E16, and E17. In this scheme, MpNs, neurons undergoing multipolar-to-bipolar transition, and bipolar neurons can be collected at E15, E16, and E17, respectively (Fig. 2A). As a result of the microarray analyses of these neurons, 2377 probes with significant changes in the expression levels were extracted (Ohtaka-Maruyama et al, 2018). We re-analyzed these data, and extracted genes that have ECM-related Gene Ontology terms. The expression levels of many ECM-related genes changed significantly during radial neuronal migration (Dataset EV1).

Among these genes, we focused on the ECM metalloproteinases as enzymes that remodel ECM structures and identified ADAMTS2 as a gene whose expression is markedly increased during radial migration. It was apparent that ADAMTS2 expression elevated more rapidly during the multipolar-to-bipolar transition than the other metalloproteinases such as ADAMTS1, ADAMTS10, MMP15, and MMP17 (Fig. 2A). ADAMTS2, ADAMTS3, and ADAMTS14 form procollagen N-proteinase subfamilies (Bekhouche and Colige, 2015; Bekhouche et al, 2016; Colige et al, 2002), and therefore, we compared their expression levels by quantitative PCR (Appendix Fig. S2). We found that the expression of ADAMTS2 was markedly upregulated during migration among the subfamily members.

In situ hybridization experiments indicated that ADAMTS2 is expressed in the SP layer (Fig. 2B). To confirm that migrating neurons express ADAMTS2, we performed double staining of the cortical sections, in which migrating neurons were labeled with GFP at E14 by in utero electroporation and fixed at E16. As a result, the expression of Adamts2 mRNA (blue) was confirmed in the GFP-positive migrating MpNs (brown) around the bottom part of the SP layer (Fig. 2C). Therefore, we decided to further analyze the function of ADAMTS2 in the regulation of neuronal migration.

## ADAMTS2 regulates neuronal migration

To investigate the involvement of ADAMTS2 in radial neuronal migration, we performed knockdown and overexpression of ADAMTS2

in migrating neurons. First, we examined the knockdown efficiency by si-RNA (Silencer Select Pre-designed si-RNA for Adamts2; s103675, Ambion) using NIH3T3 cells. As a result, 10 µg/µl of si-RNA was the most efficient in suppressing the Adamts2 mRNA expression (Appendix Fig. S3A). To see if this si-RNA can also knock down Adamts2 expression in the brain tissue, we introduced si-RNA along with GFP-expression plasmids into the neural progenitor cells by in utero electroporation at E14. Double staining of the E17 brain sections indicated that Adamts2 mRNA signals were reduced in the GFP-positive migrating neurons (Appendix Fig. S3C). Furthermore, GFP-positive knockdown neurons were isolated by FACS, and Q-PCR was performed for their RNAs. The results showed that the Adamts2 mRNA was reduced to approximately 60% of the control neurons (Appendix Fig. S3B).

We introduced si-RNA against Adamts2 and GFP-expression plasmids by in utero electroporation at E14, and the brains were fixed at E17 and E18. We quantified the distribution of GFP-positive neurons in the cortices and found that Adamts2 knockdown suppressed the radial migration, and many neurons lingered below the SP layer (Fig. 3A). We next examined the effects of Adamts2 overexpression. Notably, Adamts2 overexpression driven by both the CAG promoter and the NeuroD1 promoter inhibited migration in both cases (Figs. 3B and EV3), suggesting that Adamts2 expression levels need to be properly regulated. Knockdown and overexpression of Adamts2 did not change the distribution of Tbr2- and Ki67-positive cells, suggesting that migration defects were not due to the changes in the proportion of progenitor cells (Appendix Fig. S4A–E). We also performed immunostaining for nestin and Pax6 to examine the integrity of the radial glial scaffold and compared overexpression or knockdown vectors in electroporated and non-electroporated anterolateral brains. We found no difference in the aspect of radial glial fibers or the number of Pax6-positive cells (Appendix Fig. S5), suggesting that manipulating Adamts2 expression does not affect progenitor cell morphology or status. The migration defects of Adamts2 knockdown neurons were rescued by introducing the Adamts2-expression plasmid along with si-RNA (Fig. 3C). For the rescue experiments, we first tried three different concentrations (0.5, 1.0, and 2.0 µg/µl) of si-RNA-resistant Adamts2 expression plasmid. The results showed that 0.5 µg/µl was the most efficient for the rescue, whereas the migration defects were not rescued at 1.0 µg/µl (Appendix Fig. S6). At 2.0 µg/µl, migration was rather inhibited (Appendix Fig. S6), probably because of the excess expression of Adamts2. These results suggest that there is a suitable dose for ADAMTS2 functions, and too much or too little ADAMTS2 leads to impaired migration.

Then, we further examined Adamts2 KO mice (BJ6/ADAMTS2 Δ28KO mice) (Yamakage et al, 2019). In the heterozygous Adamts2 KO mice, most migratory neurons were stagnant below the SP

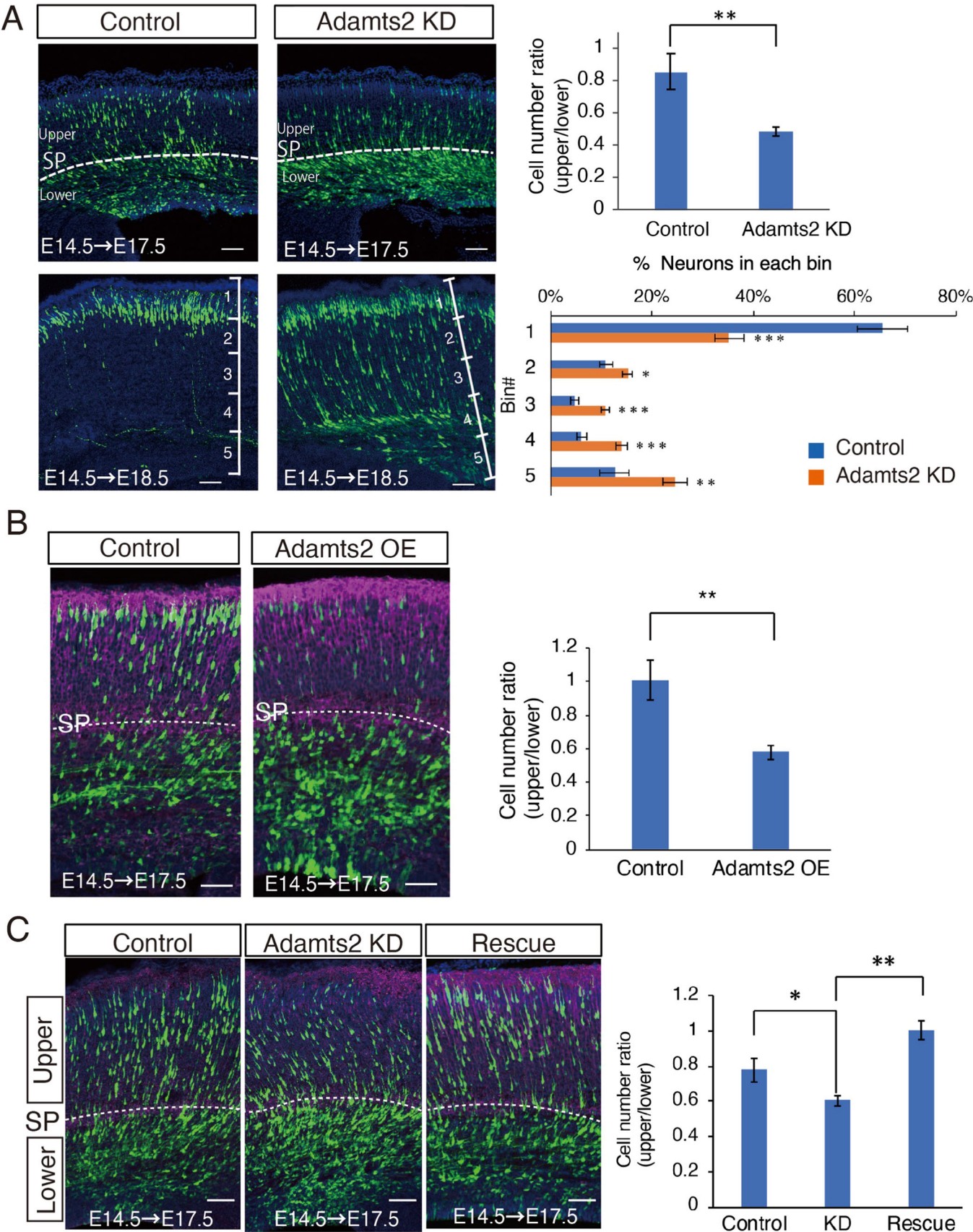

Figure 3.   *Adamts2* is involved in the regulation of radial neuronal migration.

(A) GFP-expression plasmids were co-electroporated with si-RNA for *Adamts2* at E14.5, and the distribution of GFP-positive neurons was analyzed at E17.5 and E18.5. Compared with the control, knockdown of *Adamts2* resulted in the retardation of radial neuronal migration. Many neurons were stacked just below the SP layer at E17.5 and E18.5. A part of the neurons remained in the middle of migration at E18.5 ($N = 18$ sections for each group; three fetuses from two mother mice were collected, and we used three sections from each brain for quantification. A graph quantifying the migration status of each experiment is shown on the right side of each. (B) Neuronal migration was impaired by overexpression of *Adamts2* ($N = 7$ sections for each group; two or three fetuses from three mother mice were collected, and we used one section from each brain for quantification. (C) The impairment of migration by *Adamts2* knockdown was rescued by co-electroporation of *Adamts2* expression vectors ($N = 5$ sections for each group; one or two fetuses from three mother mice were collected, and we used one section from each brain for quantification). Data information: In (A), (B), control, and si-RNA knockdown or Adamts2 overexpression brains were recovered and counted in pairs on the same litter. In (C), control, knockdown, and rescued brains were recovered from the same litter. The statistical significance for each pair was measured by unpaired, two-tailed t-tests (*$p < 0.05$; **$p < 0.01$; ***$p < 0.001$). Scale bars, 50 μm.

layer, which was similar to the phenotype of Adamts2 knockdown (Fig. EV4). These results suggested that an appropriate level of Adamts2 expression is required for radial neuronal migration. However, migration defects were not observed in the homozygous Adamts2 KO mice (Fig. EV4). This may be explained by the genetic compensation in the homozygous mice that lack the Adamts2 gene. We observed that there was a trend toward an increased expression of Adamts3 and Adamts14 in the cerebrum of homozygous Adamts2 KO mice, although they were not significantly different because of the highly variable expressions among the animals (Appendix Fig. S7). To investigate whether the other Adamts members can rescue migration defects in Adamts2 knockdown neurons, we constructed expression plasmids, in which Adamts3 and Adamts14 cDNAs were subcloned directly under the NeuroD1 promoter. They belong to the same subfamily as Adamts2. As a result, Adamts3, but not Adamts14, rescued the knockdown phenotype of Adamts2 in a neuron-specific manner (Fig. EV5A,B). These results suggest that Adamts3 is involved in the compensation mechanism in the homozygous Adamts2 KO mice.

To observe the details of the migration defect, Lifeact-GFP plasmid, which can label F-actin, was introduced into the migrating neurons along with RFP plasmid by in utero electroporation at E14.5. After two days, the brains were dissected, and the cortical slices were cultured for time-lapse observation (Fig. 4A,B, Movie EV2). In contrast to the control, where many migrating neurons changed from the multipolar to bipolar shape, many migrating neurons in Adamts2 knockdown remained multipolar and showed altered F-actin localization. When we measured the velocity vertically toward the brain surface, the migration speed was significantly reduced in Adamts2 knockdown neurons (Fig. 4C). In the control slices, F-actin was detected in the multiple processes of MpNs, but after the multipolar-to-bipolar (MP-BP) transition, F-actin was concentrated in the leading processes of bipolar neurons undergoing locomotion. On the other hand, in Adamts2 knockdown neurons, the dynamics of F-actin remained the same as in the control MpNs for at least 24 h, and F-actin did not assemble in the leading processes (arrowheads in Fig. 4B, Movie EV2). The measurement of the changes in the MP:BP ratio over the 10 h showed that the MP-BP transition process was impaired in the knockdown samples (Fig. 4C). These results indicated that ADAMTS2 is involved in the multipolar-to-bipolar transition of migrating neurons.

## TGF-β signaling-related proteins are localized at the SP layer

Recently, it was demonstrated that TGF-β signaling-related proteins, such as LTBP1 and TGF-βRIII, are potential substrates

of ADAMTS2, suggesting the participation of this protease in the control of TGF-β activity (Bekhouche et al, 2016). It was also proposed that the polarity transition of newborn neurons mechanically resembles the epithelial-mesenchymal transition that occurs during cancer cell invasion, in which TGF-β signaling is deeply involved (Singh and Solecki, 2015). Therefore, we focused on TGF-β signaling as a candidate target for ADAMTS2.

First, we examined the localization of TGF-β signaling-related proteins in the developing neocortex: phospho-Smad2/Smad3 (p-smad2/3), a major downstream effector of TGF-β signaling, tissue inhibitor of metalloproteinase 2 (Timp2), an inhibitory factor of TGF-β signaling, and TGF-βRII. Immunohistochemistry revealed that p-smad2/3 and Timp2 were localized in the lower and the upper parts of the SP layers, respectively (Fig. 5A–H). Immunoreactivity to TGF-βRII was also located in the SP layer (Fig. 5I,I').

Next, we examined the expression of CTGF in the cerebral cortices of Adamts2 KO mice by quantitative PCR. CTGF is a direct downstream target gene of TGF-β signaling (Duncan et al, 1999; Igarashi et al, 1993; Kothapalli et al, 1997) and is expressed selectively in the SP layer (Heuer et al, 2003; Hoerder-Suabedissen et al, 2009). The results showed that the expression of CTGF in the cerebral cortex was significantly reduced in homozygous KO mice compared with that of wild-type mice (Fig. 5J), suggesting that TGF-β signaling was down-regulated in the brains of Adamts2 KO mice.

## Effects of TGF-β signaling on neuronal migration

To investigate the effects of TGF-β signaling on radial neuronal migration, TGF-βRII was overexpressed by in utero electroporation at E14.5, and the brains were fixed at E17.5. The results showed that migrating neurons stagnated below the SP layer when TGF-βRII was overexpressed (Fig. 6A). Yi et al also reported that knockout of TGF-βRII in migrating neurons resulted in a delayed migration and axonal loss phenotype (Yi et al, 2010). We then analyzed the effects of the inhibitors of TGF-β signaling on the actin dynamics and migration of neurons. We prepared slice cultures of the cerebral cortex at E16.5 after introducing RFP and Lifeact-GFP-expression plasmids by in utero electroporation at E14.5. Then, TGF-βRI inhibitors (RepSox and LDN-212854) were added to the culture medium, and the movements of migrating neurons were monitored by time-lapse imaging for 3 days. In the presence of inhibitors, the neurons remained multipolar and stagnated under the SP layer, whereas many control neurons migrated across the SP layer and exhibited a bipolar shape (Fig. 6B,C). Furthermore, inhibitor-treated neurons showed abnormal dynamics of F-actin, and the

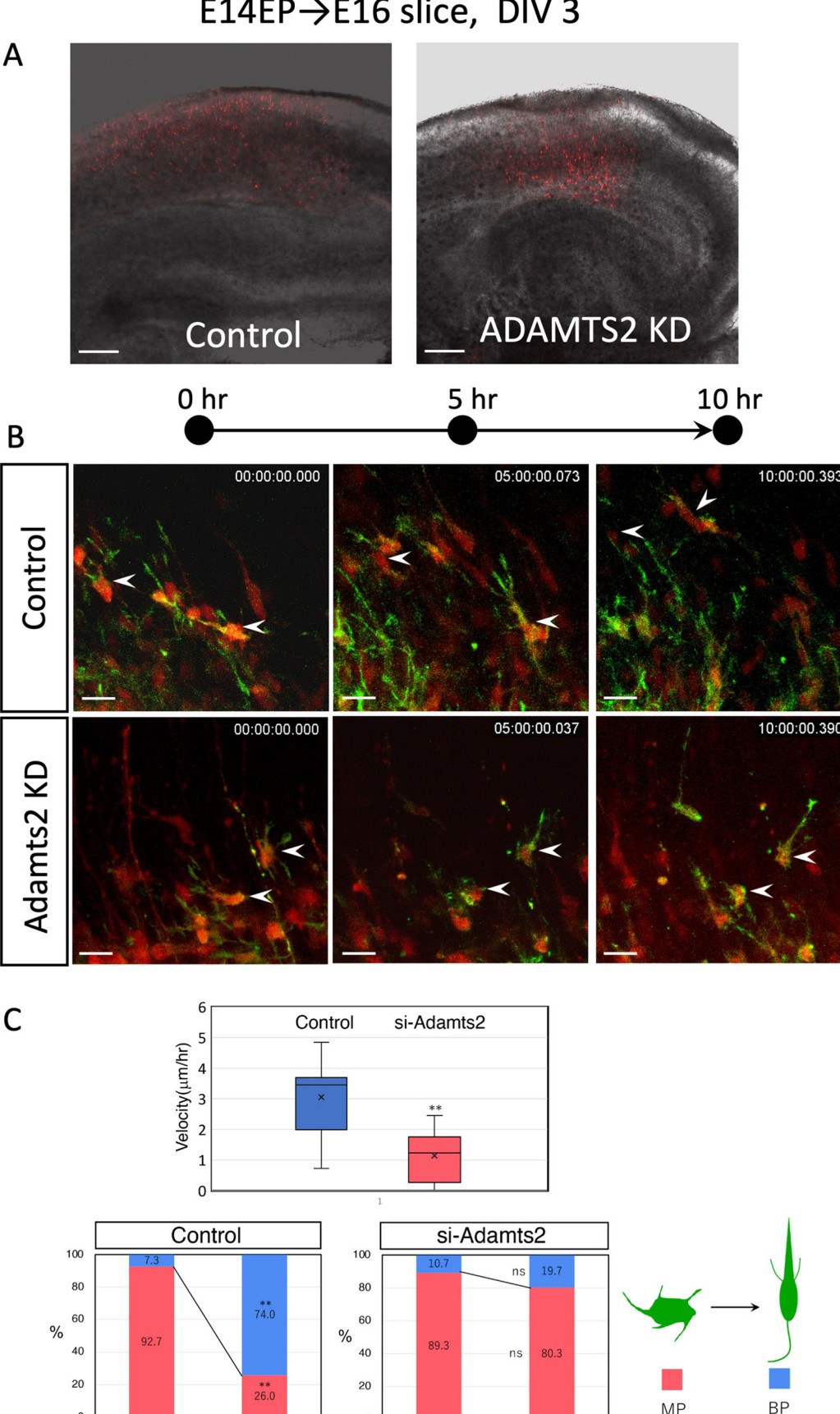

E14EP→E16 slice, DIV 3

A

Control    ADAMTS2 KD

0 hr    5 hr    10 hr

B

Control

00:00:00.000    05:00:00.073    10:00:00.393

Adamts2 KD

00:00:00.000    05:00:00.037    10:00:00.390

C

Control    si-Adamts2

Velocity(μm/hr)

**

Control

%    7.3    **
74.0

92.7    **
26.0

0hr    10hr

si-Adamts2

%    10.7    ns    19.7

89.3    ns    80.3

0hr    10hr

MP    BP

**Figure 4. Time-lapse imaging of *Adamts2* knockdown revealed that knockdown neurons are impaired in multipolar-bipolar conversion.**

(A) CAG-Lifeact (F-actin labeling) and RFP plasmids were electroporated at E14.5 and cultured slices were prepared at E16.5. Time-lapse imaging was performed for 24 h. Images of slices after 3 days in culture. Many cells were remained under the SP layer in the Adamts2-knockdown slices. (B) The localization of F-actin (Lifeact:green) was disturbed in the knockdown slices (see Movie EV2). In the control slices, MpNs transformed into bipolar neurons after 10 h (arrowheads), whereas Adamts2-knockdown neurons remained multipolar for 10 h. (C) The migration speed in the direction of the brain surface was measured (top left) ($N = 10$ cells each). The speed of Adamts2-knockdown neurons was significantly low compared with that of the control. The number of multipolar and bipolar cells was counted at the start of imaging and after 10 h ($N = 3$ slices from three independent experiments). Knockdown of Adamts2 suppressed the multipolar-to-bipolar transition (Bottom). Data information: The statistical significance for each pair was measured by unpaired, two-tailed t-tests (*$p < 0.05$; **$p < 0.01$; ***$p < 0.001$). Scale bars: (A) 100 μm; (B) 10 μm.

formation of leading processes was impaired (Movie EV3), the phenotype of which was similar to that of Adamts2 knockdown neurons (Fig. 4B, Movie EV2). Morphological analyses revealed that the inhibitor-treated neurons failed in the MP-BP transition (Fig. 6C). These results suggest that TGF-β signaling transiently activated around the SP layer is involved in the multipolar-to-bipolar transition of migrating neurons. In the inhibitor-containing slices, the axon elongation of migrating neurons was also significantly impaired, suggesting that TGF-β signaling also regulates axon determination and elongation of newborn neurons (Appendix Fig. S8) (Yi et al, 2010).

## ADAMTS2 turns on the TGF-β signaling at the SP layer and leads to morphological changes in migrating neurons

We visualized TGF-β signaling by luminescence imaging to confirm that TGF-β signaling is turned on in the migrating neurons at the SP layer. We created a construct in which two DNA sequences that are reactive to p-smad2/3 (TGF-β-reactive elements) were placed in tandem, upstream of the Emerald-Luc coding sequence that is fused with the PEST sequence, a protein-destabilizing signal (Fig. 7A). In the cells transfected with this construct, luciferase is only expressed after activation of TGF-β signaling. The plasmid was introduced into the cortical neurons by in utero electroporation, and the brain slices were observed by luminescence imaging. The luminescence signals were detected at the lower part of the SP layer and the upper part of the intermediate zone, and the signals were suppressed in the presence of RepSox, indicating that they were specific (Fig. 7C, Movie EV4). The luminescence signals were significantly reduced in the Adamts2 knockdown neurons, indicating that ADAMTS2 is required for the activation of TGF-β signaling (Fig. 7B–F, Movie EV4). The magnified time-lapse images reveals that the MpNs in the lower part of the SP layer temporarily showed a strong luminescence signal, but then the signal vanished (Fig. 7E). These results suggest that ADAMTS2 activates TGF-β signaling in the MpNs around the bottom of the SP layer, drastically reducing it after the transition from multipolar to bipolar.

If TGF-β signaling is a downstream target of Adamts2, the phenotype of impaired migration due to Adamts2 deficiency can be rescued by elevating TGF-β signaling. Thus, we tested whether overexpression of TGFβRII or TGFβ2 can rescue the phenotype in heterozygous Adamts2 KO mice by in utero electroporation using the NeuroD1 promoter (Fig. 7G). The results showed that the migration defects were rescued by overexpression of TGFβ2 but not by TGFβRII. Since the NeuroD1 promoter drives gene expression in the differentiated neurons (Guerrier et al, 2009), it is likely that TGFβ2 rescued migration around the intermediate zone and SP but not at the ventricular zone. These results strongly suggest the

presence of TGF-β signaling downstream of Adamts2, around the SP. Furthermore, to determine whether the migration defect of Adamts2 overexpression can be rescued by the knockdown of TGF-β signaling, we performed both overexpression of Adamts2 and shRNA knockdown under the NeuroD1 promoter. We constructed a plasmid in which sh-TGFβRII is expressed in a NeuroD1-Cre-dependent manner. As a result, as shown in Appendix Fig. S9, there was a trend toward rescue. Although the difference was not statistically significant, this result suggested that excess Adamts2 overactivated TGF β signaling around the SP, leading to migration defects.

## Discussion

In this study, we revealed that the SP layer is enriched with many ECM components and works as a strategically important region where TGF-β signaling is activated. ADAMTS2 is secreted by the migrating MpNs and activates TGF-β signaling in these neurons, which induces multipolar-to-bipolar transitions and locomotion.

Unlike many signaling molecules, TGF-β is sequestered in the ECM in a latent form, and it should be activated by multiple steps before binding to the receptors (Huang and Chen, 2012; Lockhart-Cairns et al, 2022; Rifkin et al, 2022; Tatti et al, 2008). The secretion and extracellular regulation of TGF-β is considered to proceed as follows. After translation, pro-TGF-β dimerizes and is then bound to LTBP by a disulfide bond in the endoplasmic reticulum. In the *trans*-Golgi network, the pro-TGF-β dimer is cleaved into a TGF-β dimer and LAP to form the small latent complex, in which LAP is non-covalently bound to TGF-β. The LLC is then secreted and sequestered in the ECM by binding to fibronectin fibers and fibrillin microfibrils until it is released by activators. Activation and release of TGF-β from the latent complex can be mediated by numerous factors such as proteinases, integrins, and some chemicals. In the SP layer, LTBP1, fibrillin 2, fibronectin, neurocan, and versican are richly expressed (Figs. 1A and EV1). The C-terminal G3 domain of versican interacts with fibronectin and fibrillin 2, and the N-terminal G1 domain interacts with hyaluronan (Wu et al, 2005). Neurocan also interacts with hyaluronan (Margolis et al, 1996). Thus, it is likely that hyaluronan, versican, neurocan, fibrillin 2, and fibronectin form a macromolecular complex in the SP layer where LLC is sequestered.

Based on this consideration, we propose the following hypothetical model for the ADAMTS2-mediated activation of TGF-β signaling (Fig. 7H). Around the bottom part of the SP layer, the migrating MpNs secrete ADAMTS2, which cleaves TGF-β-related ECM proteins such as LTBP1 and versican. After these initial cleavages, active TGF-β is released from the ECM, which

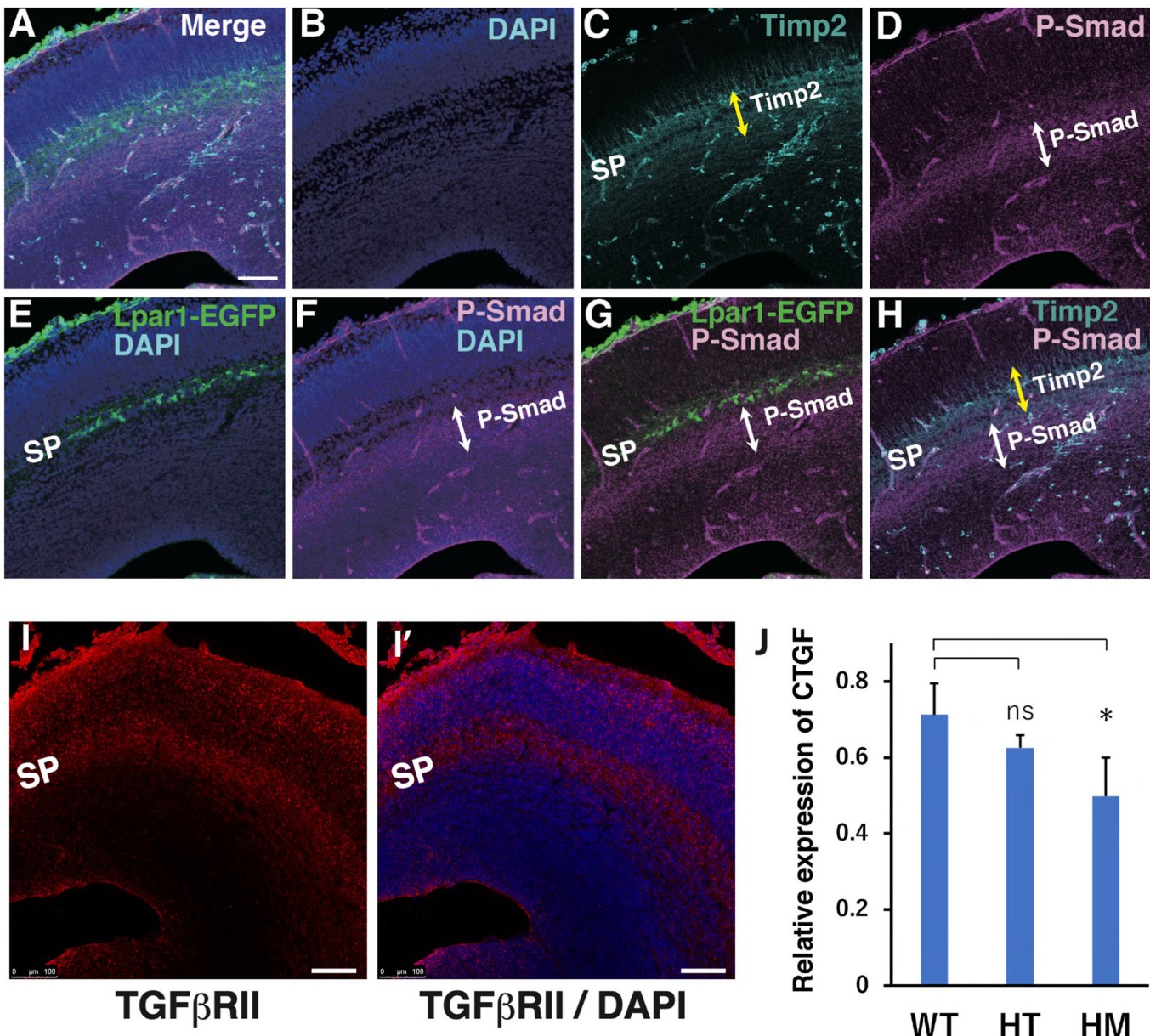

**Figure 5. TGF-β signaling-related proteins are localized at the SP layer.**

(A–H) Cortical sections from Lpar1-EGFP mice, in which SP neurons expressed EGFP, were immunostained with antibodies against TGF-β signaling-related proteins. Timp2 and p-Smad were expressed at the upper part and the lower part of the SP layer, respectively. (I, I') TGF-βRII immunoreactivities were localized at the SP layer and the upper part of the intermediate zone. (J) The expression of CTGF, a direct downstream target of TGF-β signaling, was down-regulated in the cerebral cortex of Adamts2 KO mice. The expression levels of CTGF were measured by Q-PCR using mRNAs isolated from the cerebral cortex ($N = 3$ sections for each group: three different embryos (E18) were used for collecting cerebral cortex). All sections were E15.5. Data information: The statistical significance for each pair was measured by unpaired, two-tailed t-tests (*$p < 0.05$; **$p < 0.01$; ***$p < 0.001$). Scale bars, 100 μm.

initiates the activation process of TGF-β signaling, leading to the multipolar-to-bipolar transition and switching of the migration mode (Fig. 7H). At present, we do not know the detailed structure of the ECM in the SP layer. However, CHase ABC treatment disturbed the multipolar-to-bipolar transition at the SP layer (Fig. 1C,D), suggesting that CSPGs and hyaluronan are the critical components of the SP layer ECM. Future studies are necessary to reveal the ultrastructure of the SP layer ECM, including whether fibrillin-2 and fibronectin are assembled into fiber structures. We

also do not know the activation processes of TGF-β after ECM cleavage by ADAMTS2. One possibility is that ADAMTS2 loosens the SP layer ECM, which enables the migrating neurons to access TGF-β. Alternatively, ECM cleavage by ADAMTS2 may trigger multi-step processing of LLCs, which leads to the release of active TGF-β dimer. However, the step of TGF-β activation following ECM cleavage by ADAMTS2 remains unclarified.

A previous study reported that no apparent defects were observed in the layer formation and structural maintenance of

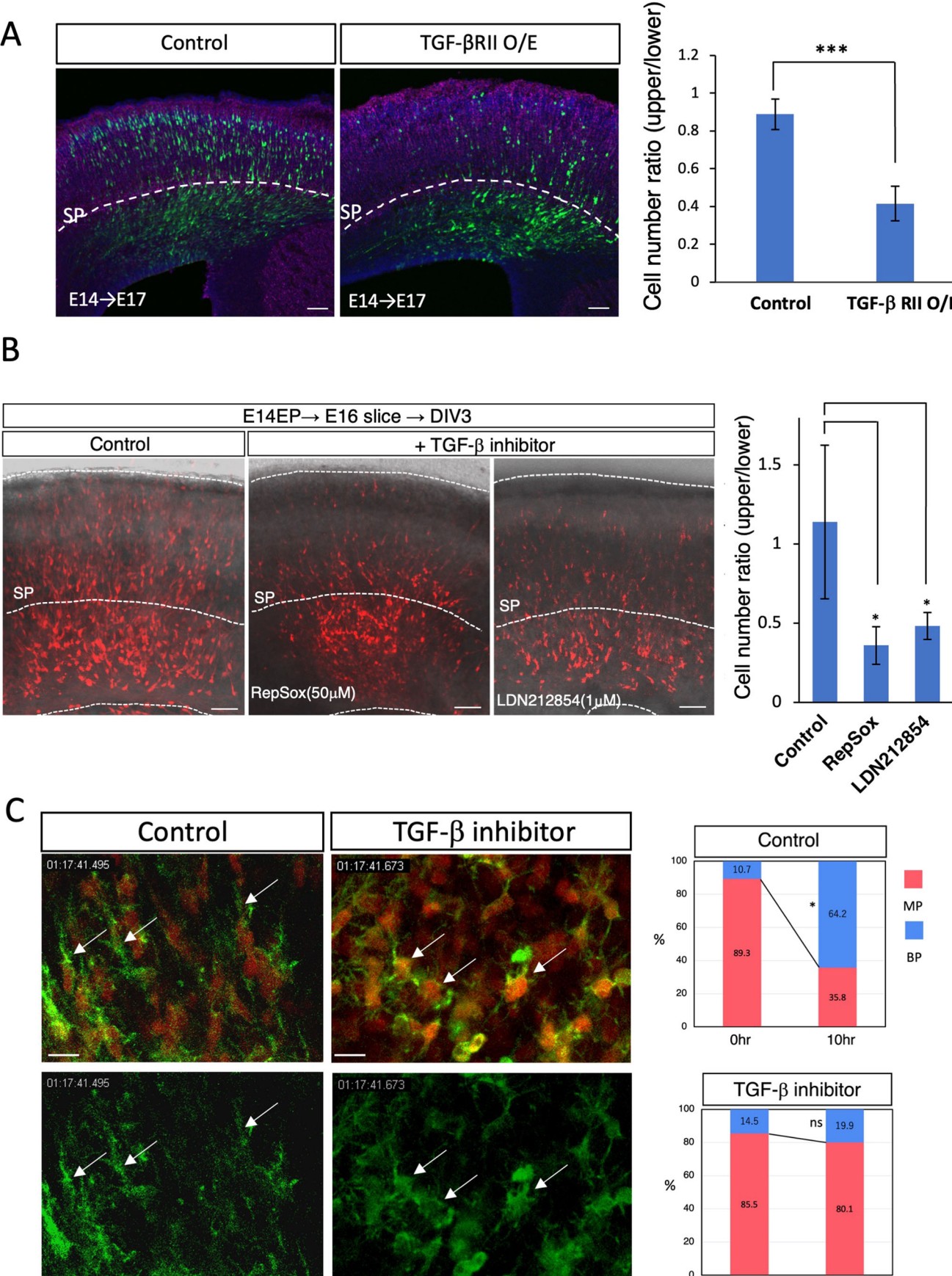

**Figure 6. Perturbation of TGF-β signaling impaired radial neuronal migration.**

(A) Overexpression of TGF-βRII in the migrating neurons impaired radial neuronal migration (N = 8 sections for each group; two fetuses from two mother mice were collected, and we used two sections from each brain for quantification. Control and overexpression were counted in pairs on the same litter). (B) Inhibitors of TGF-β receptor (50 μM RepSox and 1 μM LDN212854) disturbed radial neuronal migration in the cultured slices (N = 3 slices from three independent experiments). A graph quantifying the migration status of each experiment (A, B) is shown on the right side of each. (C) Selected images from the time-lapse recordings of F-actin dynamics shown in Movie EV3 (left). CAG-Lifeact and RFP plasmids were electroporated at E14.5. The brain slices were prepared at E16.5 and cultured in the presence or absence of 50 μM RepSox. The multipolar-to-bipolar transition was impaired in the presence of the inhibitor. Arrows indicate neurites. In control, cells became bipolar and had leading process, whereas those in the TGFβ inhibitor-treated group were observed to have multipolar neurites. Morphological analyses indicated that the multipolar-to-bipolar transition was impaired in the inhibitor-treated slices (N = 4 imaging areas from two experiments) (right). Data information: The statistical significance for each pair was measured by unpaired, two-tailed t-tests (*p < 0.05; **p < 0.01; ***p < 0.001). Scale bars: (A, B) 50 μm; (C) 10 μm.

the adult neocortex of Adamts2 KO mice (Yamakage et al, 2019). We also observed no apparent defects in the neuronal migration in the E17.5 cortices of homozygous Adamts2 KO mice, probably because of the compensatory activity by ADAMTS3, which recognize and cleave the specific sites on the substrates similar to those of ADAMTS2 (Bekhouche et al, 2016) (Fig. EV5). However, the defects of multipolar-to-bipolar transitions and radial migration were observed in the E17 cortices of heterozygous Adamts2 KO mice. Similar defects were observed in the Adamts2 knockdown neurons, and even in the Adamts2 overexpression neurons. These findings suggest that strictly regulated expression of Adamts2 is necessary to control TGF-β signaling and radial migration. Under Adamts2 knockdown and heterozygous Adamts2 KO conditions, MpNs may not be able to secrete enough ADAMTS2 to release sufficient amounts of TGF-β due to deficient ECM cleavage in the SP layer. On the contrary, Adamts2 overexpression may cause excessive cleavage of the ECM, leading to the overactivation of TGF-β signaling. Overexpression of TGF-βRII resulted in migration defects similar to the Adamts2 overexpression, suggesting that excess TGF-β signaling disturbs radial migration (Fig. 6A). In fact, the migration defects by Adamts2 overexpression tended to be rescued by knockdown of TGF-βRII (Appendix Fig. S9). Thus, the activities of TGF-β signaling in the SP layer are strictly controlled by the transient expression of an appropriate level of ADAMTS2 by MpNs.

It is well known that ADAMTS2, 3, and 14 cleave the N-terminal propeptide of fibrillar procollagen. However, in addition to having fragile skin due to abnormal collagen structures, Adamts2 KO homozygous mice have been genetically linked to male infertility (Bekhouche and Colige, 2015; Li et al, 2001). Concerning female Adamts2 KO mice, only a limited number of pups can be obtained from crosses with wild-type males, probably because of connective tissue-related problems that prevent efficient fertilization (Bekhouche and Colige, 2015). Furthermore, a genetic link between the Adamts2 gene and pediatric stroke has been suggested in humans (Arning et al, 2012). These findings suggest a sizable unknown substrate repertoire beyond procollagen N-terminal cleavage. In fact, secretome analyses of human fibroblasts using N-terminal amine isotope labeling of the substrate revealed many candidate substrates for ADAMTS2, 3, and 14 (Bekhouche et al, 2016). These include LTBP1, CTGF, fibronectin, and versican, which are richly expressed in the SP layer. Similar approaches may reveal other novel substrates for ADAMTS2 in the SP layer. Furthermore, ADAMTS2 and 3 cleave Reelin at the specific site (N-t-cleavage) and abolish its biological activity (Ogino et al, 2017; Yamakage et al, 2019). These findings show that ADAMTS2 has a wide range of functions in the developing cortex.

It is apparent that the activation of TGF-β signaling is a multi-step process that requires numerous factors such as proteinases and protease inhibitors. The immunoreactivity to p-smad2/3 was confined to the bottom part of the SP layer and the upper part of the intermediate zone (Fig. 5D). Zymography experiments also indicated that gelatinolytic activities were concentrated around the bottom part of the SP layer (Fig. EV2). These findings suggest that the activities of ECM proteinases that activate TGF-β signaling are spatially controlled in a strict manner. The immunoreactivity to Timp2 was detected at the upper part of the SP layer (Fig. 5C), and thus, this protease inhibitor may inhibit ECM proteinases, including ADAMTS2, and suppress the excess TGF-β signaling activity. Spatial and temporal regulation of many ECM components may contribute to the regulation of radial neuronal migration (Fig. 1A).

It was previously reported that TGF-β signaling forms a concentration gradient in the ventricular zone of the developing cerebral cortex and regulates early axon specification events of MpNs (Yi et al, 2010). We also observed that some of the TGF-β signaling-related molecules, such as LTBP1 and ADAMTS2, are commonly expressed in both the SP layer and the ventricular zone (Figs. 2B and EV1A). On the other hand, the immunoreactivity to p-smad2/3 was low in the ventricular zone compared with that in the SP layer (Fig. 5D). This is consistent with the report showing that early axon specification utilizes a non-smad Par6-dependent TGF-β signaling pathway (Yi et al, 2010). Thus, migrating neurons utilize the non-canonical and canonical TGF-β signaling pathways in the different cortical layers to regulate the early axon specification and multipolar-to-bipolar transition, respectively. Accordingly, our results of the perturbation experiments using TGF-β inhibitors may include the defects that occurred early in the ventricular zone. Moreover, our rescue experiments using expression plasmids with the NeuroD1 promoter indicated that ADAMTS and TGFβ work in the differentiated neurons at the intermediate and SP layers (Fig. 7G).

This study shows that the ADAMTS2 and TGF-β signaling function near the bottom part of the SP layer and are involved in the change of morphology of migrating neurons from the multipolar to bipolar shape. The SP layer is unique to mammals and plays a critical role in the developmental formation of the neocortex (Hoerder-Suabedissen and Molnar, 2015; Ohtaka-Maruyama, 2020). Our finding of the SP layer as a signaling center for the developing neocortex may aid the elucidation of multiple mechanisms underlying the evolution of the neocortex. Furthermore, failure of radial neuronal migration causes various diseases such as brain dysplasia and psychiatric disorders (Gleeson and Walsh, 2000; Ohtaka-Maruyama and Okado, 2015; Tabata and

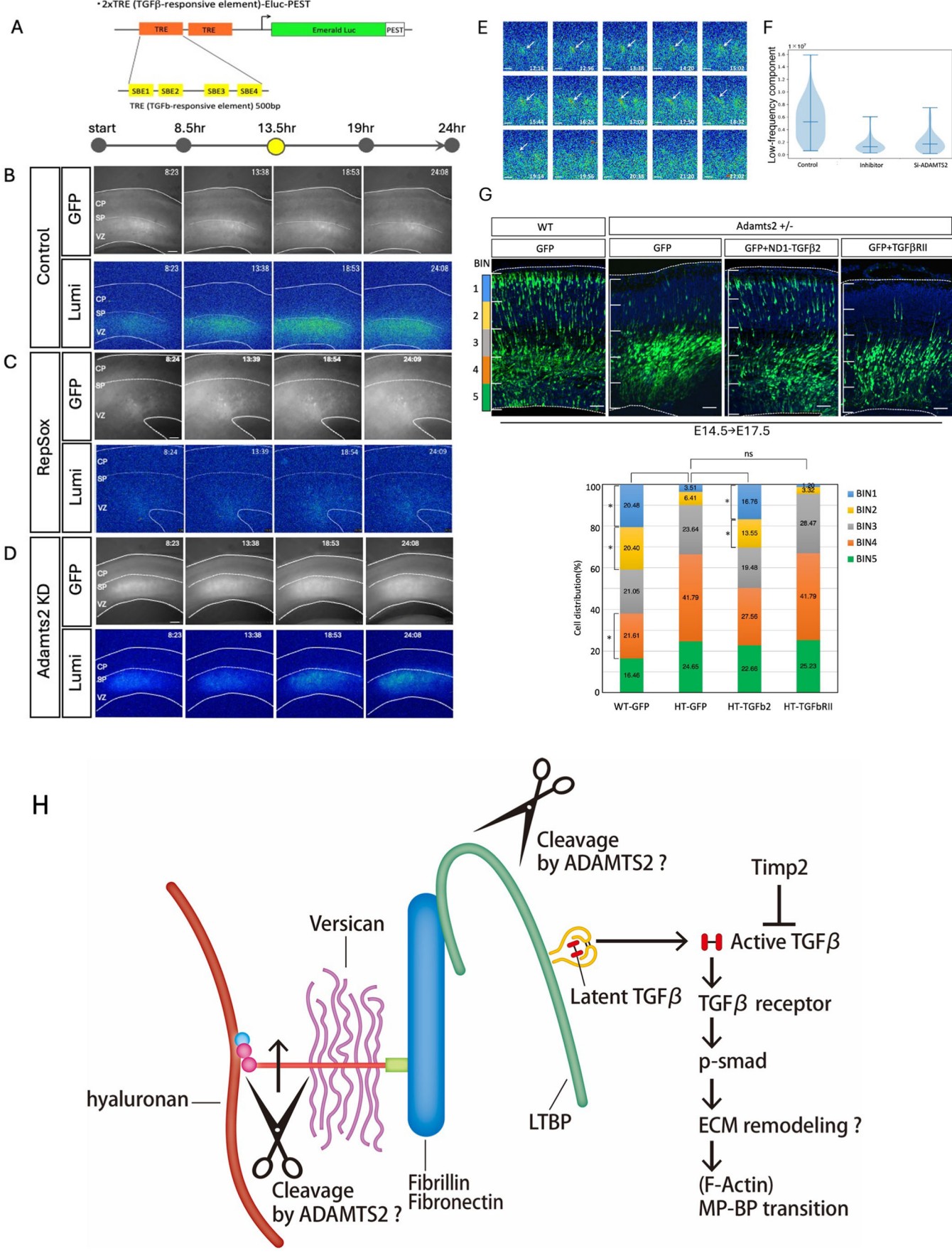

**Figure 7. Time-lapse imaging of TGF-β signaling during radial neuronal migration.**

(A) The plasmid construct used for luminescent imaging of TGF-β signaling. TGF-β-responsive elements were tandemly inserted into the upstream of Emerald Luc conjugated with the PEST sequence. (B–D) 2xTRE-Eluc-Pest plasmids were electroporated at E14.5 along with GFP-expression plasmids. The luminescence imaging was performed using cultured slices prepared at E16.5. Compared with the control (B), the luminescence signals were diminished when 50 μM RepSox was added (C), or Adamts2 si-RNA was co-electroporated (D). (E) Enlarged images revealed that a control migrating neuron transiently showed strong luminescence emission during the multipolar-to-bipolar transition from 12 to 18 h. Arrows indicate migrating neurons with positive luminescent signals. (F) Quantification of the luminescence signals ($N = 3$ slices from three independent experiments). The data were taken 4.5 h after the start of imaging. (G) Overexpression of TGF-β2 rescues the migration impairment phenotype of the Adamts2 KO heterozygous mouse cortex. Plasmids expressing TGF-β2 or TGF-βRII under the NeuroD1 promoter were used to determine if they could rescue the phenotype of impaired radial migration in the Adamts2 KO heterozygous mouse embryonic cortex. The phenotype was rescued by overexpression of TGF-β2 but not by overexpression of TGF-βRII. In utero electroporation was performed at E14.5, and the brains were fixed at E17.5. Compared with heterozygous brains in which GFP was electroporated (HT-GFP), the proportion of neurons distributed in BIN 1 and 2 significantly increased in the TGF-β2 rescued brains (HT-TGFb2) ($N = 6$ sections for each group. Three sections from two brains each were analyzed). A graph quantifying the migration status is shown at the bottom. (H) Hypothetical model for the functions of ADAMTS2. The migrating multipolar neurons transiently secrete ADAMTS2 around the bottom part of the SP layer, which cleaves TGF-β-related ECM proteins such as LTBP1 and versican. After these initial cleavages, active TGF-β- is released from the ECM, which initiates the activation process of TGF-β signaling, leading to the multipolar-to-bipolar transition and switching of the migration mode. Data information: The statistical significance for each pair was measured by unpaired, two-tailed t-tests (*$p < 0.05$; **$p < 0.01$; ***$p < 0.001$). Scale bars: (E) 10 μm; (G) 50 μm.

Nagata, 2016), and the results of this study will contribute to understanding the pathogenesis of these diseases.

## Methods

### Experimental animals

All animals were treated in accordance with the Tokyo Metropolitan Institute of Medical Science Animals Care and Use Committee guidelines. Pregnant ICR mice were purchased from Japan SLC and used for in utero electroporation and microarray analyses. Lpar1-EGFP mice (Tg(Lpar1-EGFP)GX193Gsat) were obtained from MMRRC. ADAMTS2 KO mice (BJ6/ADAMTS2 Δ28) that contain a deletion of 28 bp downstream of the start codon of exon 1 were generated using the CRISPR/Cas9 system as previously reported (Ito et al, 1995).

### Antibodies

The primary antibodies used for immunostaining were mouse anti-neurocan (Oohira et al, 1994), rabbit anti-cleaved versican (ab19345, Abcam), rabbit anti-Fibrillin-2 (bs12166R, Bioss), rabbit anti-LTBP1 (ab78294, Abcam), chicken anti-GFP (ab13970, Abcam), rabbit anti-MAP2 (AB5622, Merck Millipore), mouse anti-TIMP2 (ab1828, Abcam), rabbit anti-pSmad3 (ab52903, Abcam), and rabbit anti-TGF-βRII (bs0117R, Bioss), mouse anti-CS monoclonal antibody, CS-56 (C8305, SIGMA), rabbit anti-Ki67 (Novocastra), and rat anti-Tbr2 (SIGMA), Nestin (SC-33677, Santa Cruz). The secondary antibodies used were Alexa Fluor 488-conjugated donkey anti-chicken IgY (IgG) (703-545-155, Jackson ImmunoResearch), Alexa Fluor 546-conjugated donkey anti-rabbit IgG (A10040, Thermo Fisher Scientific), and Cy5-conjugated donkey anti-mouse IgG (715-175-150, Jackson ImmunoResearch). For double staining of in situ hybridization (Adamts2 mRNA) and immunohistochemistry (GFP), biotin-conjugated goat anti-chicken IgY (ab97133, Abcam) and streptavidin-conjugated HRP (SA-5004, Vector) were used. The antibodies were used at a 1:500 dilution unless otherwise noted.

### RT-PCR for versican

RT reactions were performed using 0.5 μg each of RNA purified from the cortex at stages E15, 16, and 17 to make 1st strand cDNA.

1 μl of template in 25 μl of reaction solution was used to perform PCR reactions for 25 cycles at an annealing temperature of 60 °C to confirm the PCR products. The primer sequences for V0, V1, and V2 were the same as those used in the paper by Asano et al, 2017 (Asano et al, 2017).

### Plasmid construction

All the cDNA fragments described below were cloned into the EcoRI sites of the pCAG-GS expression vector. pCAG-tRFP was generated by cloning the turboRFP (Evrogen) coding sequence into pCAG-GS. For the construction of pCAG-Adamts2 and pCAG-TGFβRII, the Adamts2 or TGFβRII coding sequence was amplified by PCR using mouse *Adamts2* and *TGFβRII* ORF plasmids (pCMV-*Adamts2*, Harvard plasmid clones; pCMV-*TGFβRII*, Addgene) as templates. Amplified cDNA fragments were inserted into the EcoRI sites of pCAG-GS plasmid. pCAGGS-EGFP was a gift from Dr. Ayano Kawaguchi.

Si-RNA-resistant plasmid for *Adamts2* (CAG-mr-*Adamts2*) was constructed as follows. AccIII-NcoI fragments of the Adamts2 coding sequence that contains the si-RNA target site were cut out and replaced by synthetic DNA fragments with inserted codon mutations that do not change the amino acids. pCAG-Lifeact plasmid was constructed from pEGFP-C1Lifeact-EGFP (Addgene). NheI-BglII fragments of the coding region were subcloned into the CAG-GS vector. TGFβ signal monitoring plasmid, p2XTRE-Eluc-PEST, was constructed with the pEluc (PEST)-test (TOYOBO). Four TGFβ-responsive elements (SBE) are inserted in tandem, upstream of the Emerald-Luc coding sequence. For the construction of Adamts3 and Adamts14 expression vectors, cDNA fragments of Adamts3 and Adamts14 (3.6 kb) were cloned into the EcoRI site of NeuroD1-IRES-EGFP (Addgene). For the construction of NeuroD1-TGFβ2 expression vectors, we used the TGFβ2 expression plasmid purchased from Origene (MD, USA). A 1.7 kb TGFβ-2 cDNA fragment was ligated to the EcoRI site of the NeuroD1-IRES-EGFP plasmid after blunting treatment. For the construction of NeuroD1-TGFβRII expression vectors, a 3.8 kb TGFβRII cDNA fragment obtained from the CAG-TGFβRII plasmid we have constructed above was subcloned into the EcoRI site of pNeuroD1-IRES-EGFP. NeuroD1-Cre was constructed with pNeuroD1-IRES-EGFP, for which the EGFP region was replaced with Cre. NeuroD1-LSL-shTGFβRII was constructed with pGIZ shRNA Tgfbr2 (Dharmacon) and CAG-LSL-tdTomato (TK270). A

1.6 kb EcoRI-HindIII fragment, including td-Tomato cDNA, was removed from TK270, and the rest of the vector region, including loxP-stop-loxP region, was ligated with the fragment including turboGFP-mir30a and loop region of pGIPZ shRNA Tgfbr2 plasmid using an In-Fusion HD cloning kit (Takara).

## In utero electroporation

The pregnant mice were deeply anesthetized with sodium pentobarbital at 50 mg/kg, and the uterine horns were exposed. A plasmid DNA solution (3–5 µg/µl) in HEPES buffered saline, pH 7.2 (HBS) containing 0.01% Fast Green, was injected into the lateral ventricle with a glass micropipette using a microinjector IM-31 (Narishige). Approximately 1–2 µl of the plasmid solutions were injected into E14.5 brains. The heads of E14.5 embryos in the uterus were placed between a tweezer-type electrode, 5 mm in diameter (LF650P5, BEX), and then five electric pulses (35 V, 50 ms in duration at intervals of 950 ms) were delivered using a CUY21E electroporator (BEX). For E10.5 embryos, four electric pulses (50 V, 50 ms in duration at intervals of 950 ms) were applied using a 1 mm diameter disk electrode (LF650P1). After electroporation, the uterine horns were returned to the abdominal cavity to allow the embryos to continue development.

## Immunohistochemical staining

The embryonic brains were dissected and fixed in 4% paraformaldehyde (PFA)/PBS overnight at 4 °C. The tissues were cryoprotected in 15% sucrose/PBS for 2–3 h, followed by 30% sucrose/PBS overnight at 4 °C. The brains were then embedded in OCT compound (Tissue Tek) and cut into 20-µm-thick sections using a cryostat HYRAX C50 (Zeiss). The sections were soaked in PBS for 5 min and pre-incubated with 0.01% Triton X-100/PBS for 15 min, which was then incubated overnight at 4 °C with primary antibodies diluted with PBS containing 0.5% skim milk. After washing three times with PBS, the sections were incubated with species-specific anti-IgG antibodies conjugated to Alexa Fluor 488, Alexa Fluor 546, or Cy5. Then, sections were mounted with PermaFluor (Thermo Scientific) after DAPI staining (5 µg/ml, Sigma-Aldrich). Images were captured using the Zeiss LSM710, LSM780, and Leica SP8 confocal microscopes.

## Slice cultures

Embryonic brains electroporated with various expression constructs were dissected at E15.5 or E16.5 and embedded in 3% low-melting agarose gels prepared in HBS. Embedded brains were cut into 300-µm-thick coronal slices using a vibratome VT1200S (Leica). The slices were placed on the insert membrane (PICMORG50, Merck Millipore) and then incubated in Neurobasal medium (Gibco) supplemented with B27 (Gibco) and antibiotics (Antibiotic-Antimycotic, Gibco) under 5% $CO_2$ and 60% $O_2$.

## Chondroitinase ABC treatment

The stock solution, ChondroitinaseABC (10 U/ml) (Seikagaku Corporation), was first diluted 10-fold and then 1/2000 of it was added to the medium to a final concentration of 0.5 mU/ml in the culture medium (Neurobasal supplemented with B-27).

## In situ zymography

The EnzCheckTM Gelatinase/Collagenase Assay Kit (Thermo Fisher E12055) was used for in situ zymography. Cortical slices were prepared with E17 embryonic cortices and cultured at 37 °C under 5% $CO_2$. DQ-gelatin was diluted to a 100 µg/ml final concentration with the reaction buffer (0.05 M Tris-HCl, 0.15 M NaCl, 5 mM $CaCl_2$, 0.2 mM sodium azide, pH 7.6) and incubated at 37 °C for 60 min with or without inhibitors (Fig. 1E). 1,10-phenanthroline (Sigma131377) and GM6001 (SelleckS7157) were added to a final concentration of 10 mM and 50 µm, respectively. For time-lapse imaging, RFP-expressing plasmids were electroporated in utero at E14.5, and cultured slices were prepared at E17.5. Time-lapse imaging was performed after incubating the prepared slices with DQ-gelatin solution for 30 min. After image acquisition was started, time-lapse imaging was performed every 10 min for ~16 h (Fig. EV2).

## Time-lapse imaging

The slices were cultured using stage top incubators Chamlide TC (Live Cell Instrument) for SP5 and STXG-GSI2X (TOKAI HIT) for SP8 under 5% $CO_2$ and 60% $O_2$. Time-lapse recordings were performed using a Leica SP5 or SP8 inverted confocal microscope with a 20× long-operation objective lens (HC PL FLUOTAR, L 20x/0.40 CORR, Leica). The maximum intensity projection was generated from 10–15 Z-stack images with 10-µm intervals at each time point.

## Si-RNA mediated knockdown

Silencer Select Pre-designed si-RNA (Ambion) for Adamts2 (s103675) and negative control (AM4635) were used for knockdown experiments. Si-RNAs were introduced to embryonic ventricles of the cerebral cortex by in utero electroporation along with GFP-expressing plasmids.

## In situ hybridization

RNA probe for *Adamts2* mRNA was prepared with the 1.3 kb BamH1-EcoNI fragment of the *Adamts2* coding sequence. The fragment was subcloned into pBluescriptKS+ and riboprobe was transcribed with digoxigenin-labeled dUTP. In situ hybridization and double staining with in situ hybridization and immunohistochemistry were performed as previously described (Ohtaka-Maruyama et al, 2007). The primary antibody used in this study was anti-GFP (1:500) (ab13970, Abcam).

## Q-PCR

Q-PCR was performed as described previously (Ohtaka-Maruyama et al, 2007), where 2 µg of total RNA was used to make 1st strand cDNA. The mRNA levels were quantified by real-time PCR using the ABI 7500 real-time PCR system.

## TGF-β inhibitors

The TGF-β inhibitors used in this study were RepSox (E616452, Selleck) and LDN-212854 (S7147, Selleck).

## Luminescent imaging of TGF-β signaling

The Live Cell Bioluminescence Imaging System Cellgraph AB-3000B (ATTO) was used for bioluminescence imaging. TGF-β signaling monitor plasmid, 2xTRE-Eluc-Pest, was electroporated into the mouse cortex at E14, and cultured slices were prepared at E16.5. D-Luciferin potassium salt (Fuji film) was added to the culture medium at the final concentration of 5 mM.

## Quantification of luminance values

The data analysis was performed using Python 3.8.0. The luminescence image was analyzed in a grid of $16 \times 14$ squares, and the average luminance value of each square was calculated. The average of the signals along the time axis of each grid was Fourier transformed. We used these values to visualize changes in luminance over time. A low-pass filter was applied by adding frequencies of 30 to 75 Hz for each grid, to remove high outliers caused by cosmic rays. The location of the SP layer was then confirmed in the brightfield image, and luminance values were obtained for only the top and bottom four rows of the SP layer. Because of the high background obtained immediately after the start, the first 4.5 h were excluded from the analysis. The frequency domain was then summed from 2 to 30 Hz for each grid, and the results were represented in a violin plot.

## Statistical analysis

Statistical analyses were performed using Graph Pad Prism 6.0. All data were expressed as the mean ± SEM, and Student's *t*-tests (unpaired two-tailed) were used to compare the means of the two groups.

# Data availability

The microarray data of migrating neurons are the same as those used in the previously published paper (Ohtaka-Maruyama et al, 2018) and are available in the GEO database under accession number GSE102911.

The source data of this paper are collected in the following database record: biostudies:S-SCDT-10_1038-S44319-024-00174-x.

# Peer review information

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

## Acknowledgements

We thank Kuniko Kohyama (Child Brain Project) for her advice on in situ zymography with DQ-gelatin, Song Xianghe (Neural Development Project) for analyzing data, and Aiko Odajima for her technical support in image processing. We thank members of the Neural Development Project of Tokyo Metropol. Inst. Med. Sci. for discussion and comments on the study. This work was supported in part by the JSPS KAKENHI-Grants (17K07428, 19H04795, 20H03270 to CO-M, and 16K07077 to NM), and AMED under Grant Number JP21gm1310012, Y2018 Research Grant from Takeda Science Foundation, The Naito Foundation, FY2020 Research grant from the Novartis Foundation, Brain Science Foundation, FY2021 Research Grant from Yamada Science Foundation, KOSE Cosmetology Research Foundation, The Mitsubishi Foundation, Astellas Foundation for Research on Metabolic Disorders to CO-M.

## Author contributions

**Noe Kaneko**: Investigation. **Kumiko Hirai**: Investigation. **Minori Oshima**: Data curation. **Kei Yura**: Supervision. **Mitsuharu Hattori**: Resources; Supervision. **Nobuaki Maeda**: Supervision; Funding acquisition; Writing—original draft. **Chiaki Ohtaka-Maruyama**: Conceptualization; Data curation; Formal analysis; Supervision; Funding acquisition; Investigation; Writing—original draft; Project administration; Writing—review and editing.

Source data underlying figure panels in this paper may have individual authorship assigned. Where available, figure panel/source data authorship is listed in the following database record: biostudies:S-SCDT-10_1038-S44319-024-00174-x.

## Disclosure and competing interests statement

The authors declare no competing interests.

# Expanded View Figures

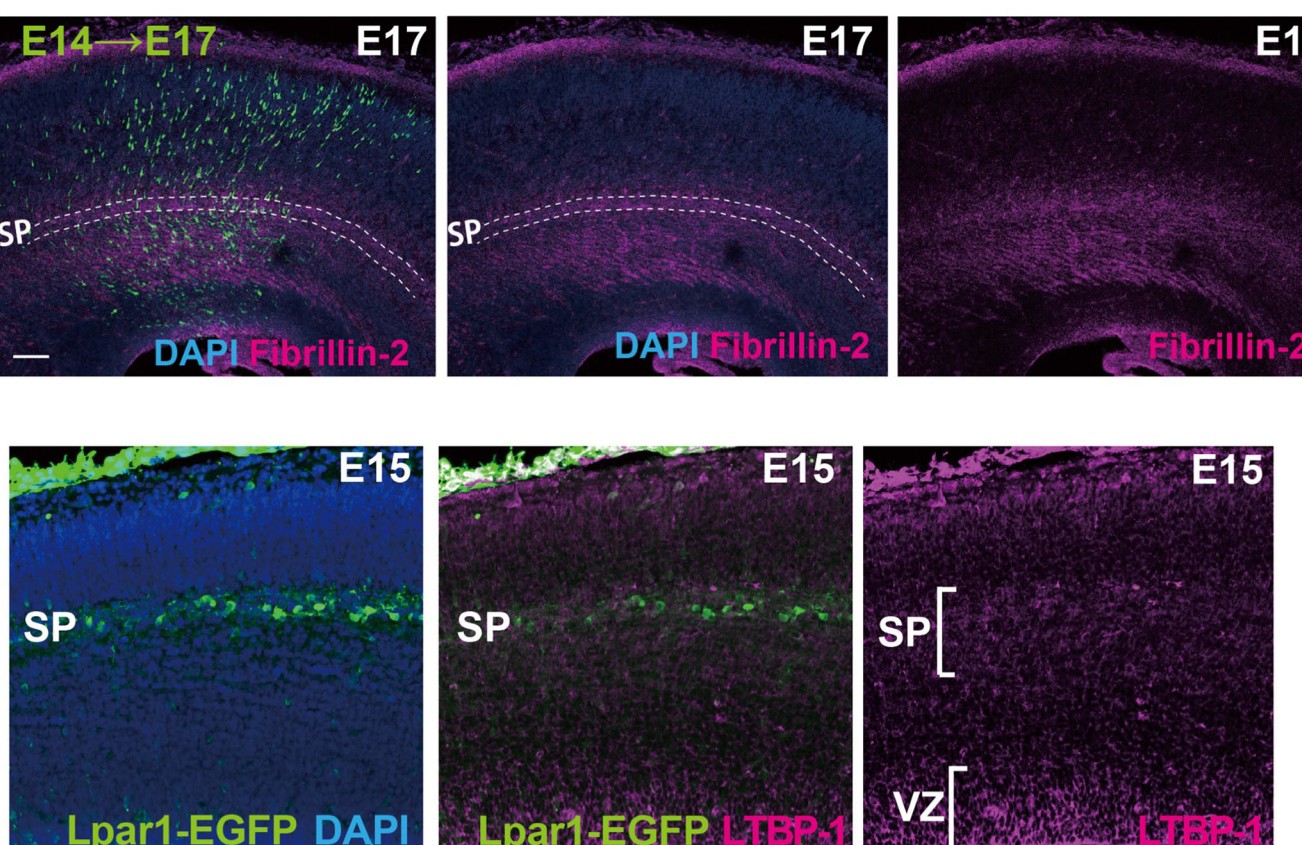

**Figure EV1. The subplate layer is rich in ECM components.**

(A) In situ hybridization databases revealed that mRNAs of genes encoding ECM proteins are localized at the subplate layer in the developing mouse cortex (arrowheads). The data for Fibronectin 1, Collagen XIa1, and PTPRZ1 are from Allen brain atlas and the data for LTBP1 is from Gene Paint. (B) Immunohistochemistry of Fibrillin-2 indicated that this protein is localized at subplate (SP) and intermediate zone. The migrating neurons were labeled with EGFP by in utero electroporation at E14. The immunoreactivities for LTBP-1 were localized at SP and ventricular zone (VZ). In this section, SpNs were labeled by EGFP.

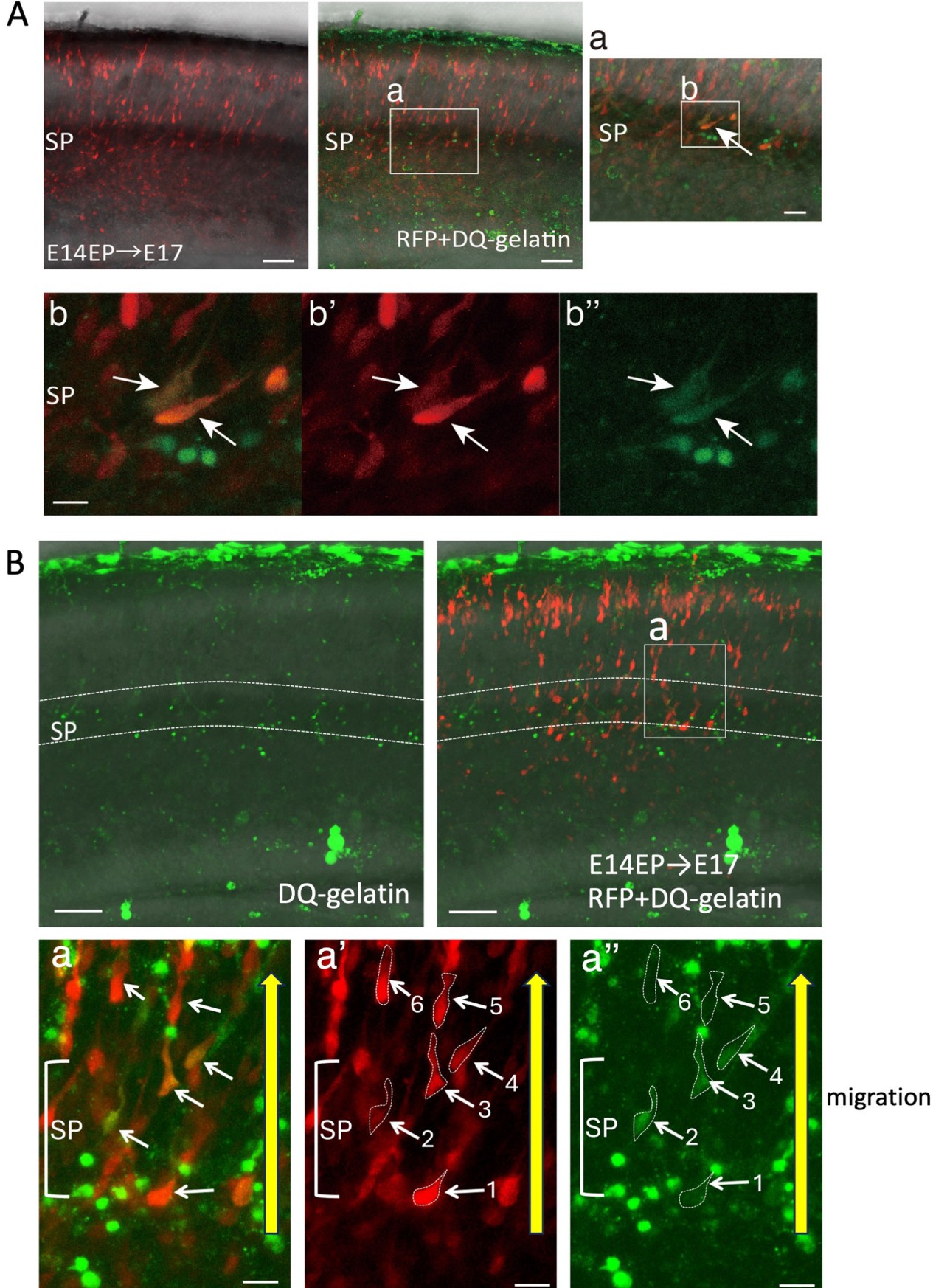

◀   **Figure EV2.  In situ zymography using DQ-gelatin revealed that ECM protease activity (green) occurred in the SP layer.**

(**A**) RFP expression plasmids were electroporated in utero at E14.5, and brains were dissected at E17.5. Cultured slices were prepared using these electroporated brains and were incubated with DQ-gelatin. Image acquisition began 30 min after incubation, and time-lapse imaging was performed every 10 min for ~16 h. Enlarged images reveal that the migrating neurons exhibited gelatinolytic activities near the SP layer (indicated by arrows in a–b"). (**B**) Example of another slice. When the region (a) is enlarged, cells before entering the SP (cell 1) are green-negative, but cells entering the SP (cells 2, 3 and 4) are green-positive and show gelatinase activity. However, cells that have passed through the SP layer (cells 5, 6) are also green-negative. Scale bars, 50 μm for (**A**), (**B**), 10 μm for (**A**-a-b") and (**B**-a-a").

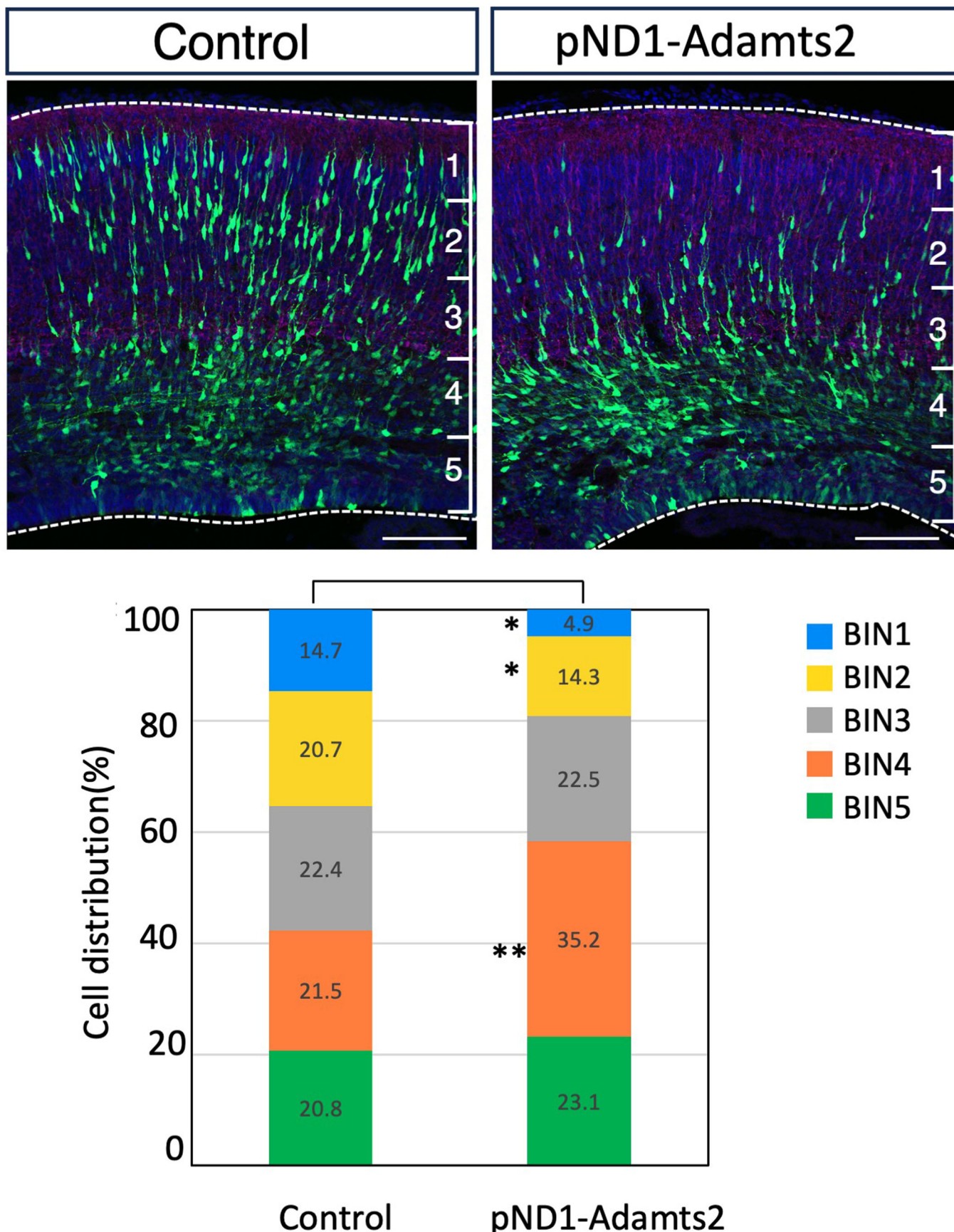

◄ **Figure EV3. Neuron-specific overexpression of Adamts2.**

Adamts2 cDNA was subcloned under the NeuroD1 promoter and used for in utero electroporation experiments. In utero electroporation was performed at E14, and the brains were dissected at E17. The number of cells distributed in the five bins was counted. Neuron-specific overexpression of Adamts2 resulted in impaired migration ($N = 5$ sections for each group; two fetuses from two mother mice were collected, and we used one or two sections from each brain for quantification. Control and overexpression were counted in pairs on the same litter) The statistical significance for each pair of the same bin was measured by unpaired, two-tailed t-tests (*$p < 0.05$; **$p < 0.01$; ***$p < 0.001$) Scale bars; 100 μm.

## E14→E17

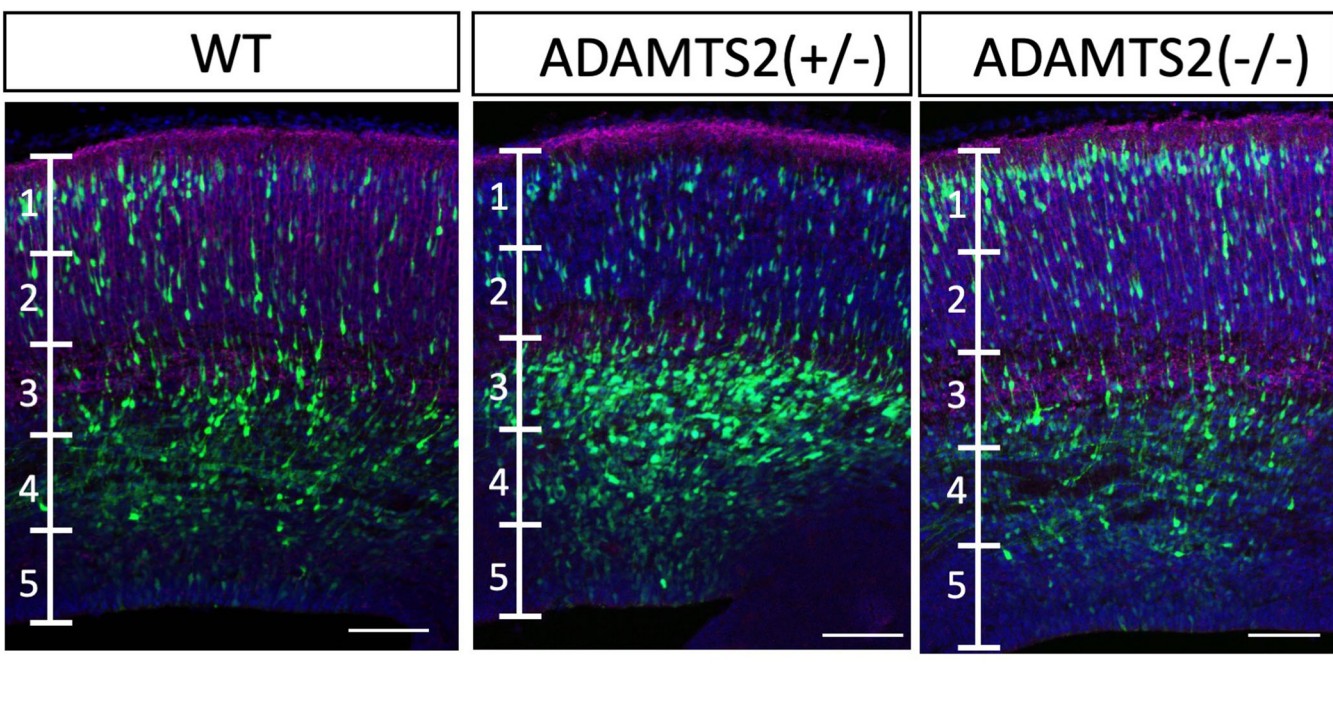

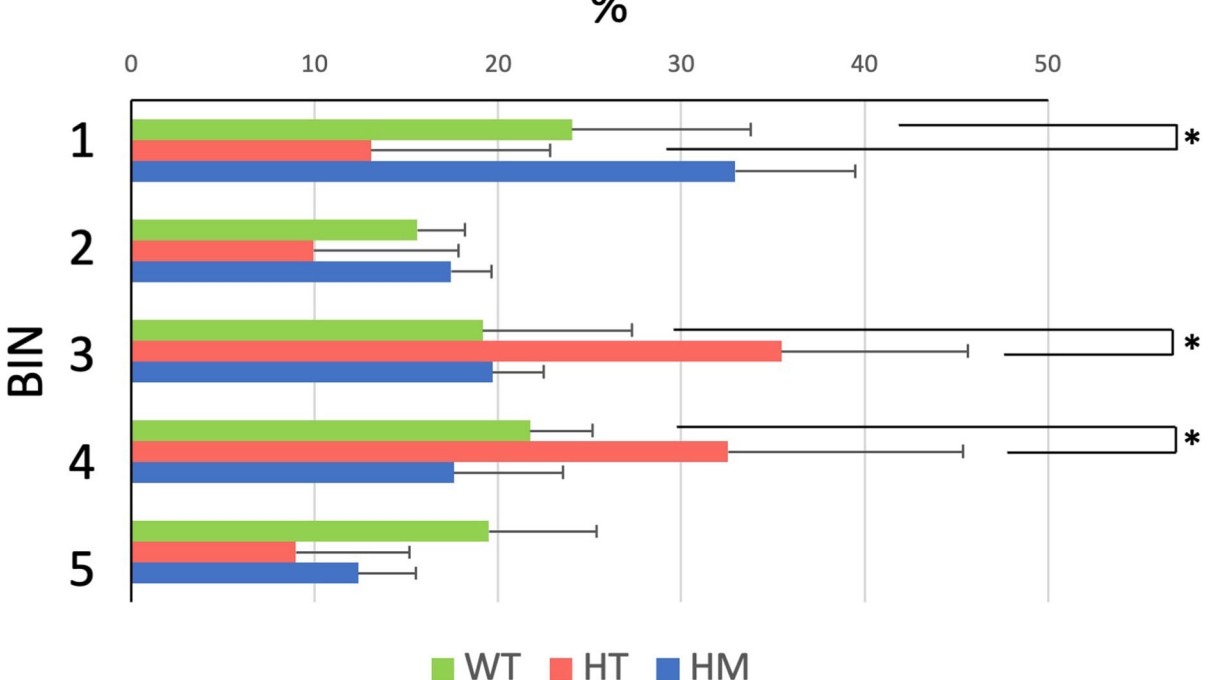

**Figure EV4.  Impaired radial neuronal migration in the brain of Adamts2 knockout heterozygous mice.**

Adamts2 KO mice showed significant migration defects in heterozygous (HT) mice, but not in homozygous (HM) mice. $N = 6$ sections; two fetuses from three mother mice for WT and HT, $N = 4$ sections; two fetuses from two mother mice for HM. The statistical significance for each pair of the same bin was measured by unpaired, two-tailed t-tests (*$p < 0.05$; **$p < 0.01$; ***$p < 0.001$). Scale bars; 100 μm.

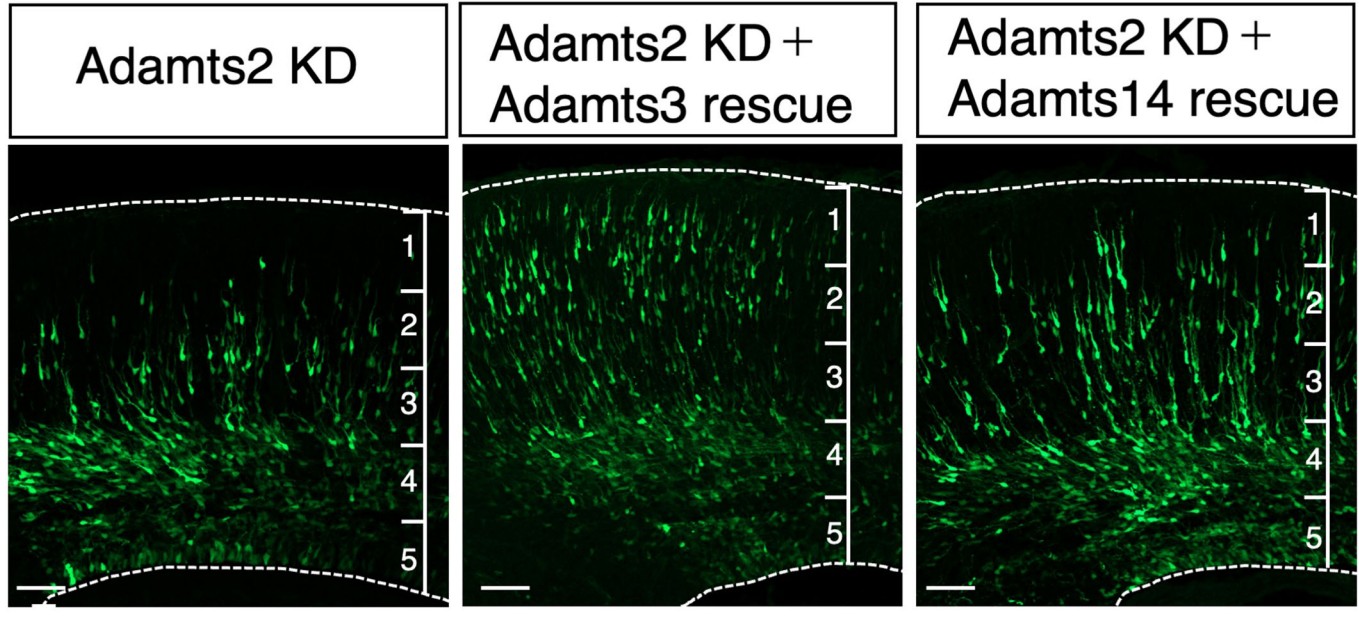

◄ **Figure EV5. Adamts3 rescued the migration defects caused by Adamts2 knockdown.**

Adamts3 and Adamts14 cDNAs were subcloned directly under the NeuroD1 promoter and used for the rescue experiments of Adamts2 knockdown. In utero electroporation was performed at E14, and the brains were dissected at E17 (**A**). The number of cells distributed in the five bins was counted (**B**). When Adamts3-expression plasmid was introduced with Adamts2 si-RNA, the migration phenotype was rescued. In the case of Adamts14, the migration phenotype was not rescued. $N = 5$–7 slices from three brains to six brains collected from two mother mice were used for the analysis. The statistical significance for each pair of the same bin was measured by unpaired, two-tailed t-tests (*$p < 0.05$; **$p < 0.01$; ***$p < 0.001$). Scale bars, 50 mm.

