## [Peer Review File · EMBO Reports]

ADAMTS2 promotes radial migration by activating TGF- β signaling in the developing neocortex

Noe Kaneko, Kumiko Hirai, Minoru Oshima, Kei Yura, Mitsuharu Hattori, Nobuaki Maeda, and Chiaki Ohtaka-Maruyama

Corresponding author(s): Chiaki Ohtaka-Maruyama (maruyama-ck@igakuken.or.jp)

Review Timeline:

Submission Date:	19th Aug 22
Editorial Decision:	20th Sep 22
Revision Received:	13th Apr 23
Editorial Decision:	7th Jun 23
Revision Received:	4th Apr 24
Editorial Decision:	16th May 24
Revision Received:	20th May 24
Accepted:	3rd Jun 24

Editor: Esther Schnapp

Transaction Report:

Dear Dr. Ohtaka-Marurama,

Thank you for the submission of your manuscript to EMBO reports. We have now received the full set of referee reports that is pasted below.

As you will see, the referees acknowledge that the findings are potentially interesting. However, they also point out that the data as they stand now are not sufficiently strong and convincing, and that significant revisions will be required before the study can be considered for publication here. They suggest several experiments to strengthen the study, and I think all suggestions are in principle good. I wanted to know whether you think that you can address all comments, especially the suggested rescue experiments with either other ADAMTS proteins or, more importantly, with TGFbeta pathway components/agonists? Also the neuron-specific ADAMTS2 knockdown experiment seems to be a good addition. Please let me know what you think. We can also discuss this in a video chat, if you prefer.

I would thus in principle like to invite you to revise your manuscript with the understanding that the referee concerns must be fully addressed and their suggestions taken on board. Please address all referee concerns in a complete point-by-point response. Acceptance of the manuscript will depend on a positive outcome of a second round of review. It is EMBO reports policy to allow a single round of major revision only and acceptance or rejection of the manuscript will therefore depend on the completeness of your responses included in the next, final version of the manuscript.

We realize that it is difficult to revise to a specific deadline. In the interest of protecting the conceptual advance provided by the work, we recommend a revision within 3 months (21st Dec 2022). Please discuss the revision progress ahead of this time with the editor if you require more time to complete the revisions.

- 1) A data availability section providing access to data deposited in public databases is missing. If you have not deposited any data, please add a sentence to the data availability section that explains that.
- 2) Your manuscript contains statistics and error bars based on $n=2$. Please use scatter blots in these cases. No statistics should be calculated if $n=2$.

3) We replaced Supplementary Information with Expanded View (EV) Figures and Tables that are collapsible/expandable online. A maximum of 5 EV Figures can be typeset. EV Figures should be cited as 'Figure EV1, Figure EV2' etc... in the text and their respective legends should be included in the main text after the legends of regular figures.

5) a complete author checklist, which you can download from our author guidelines <https://www.embopress.org/page/journal/14693178/authorguide>. Please insert information in the checklist that is also reflected in the manuscript. The completed author checklist will also be part of the RPF.

6) Please note that all corresponding authors are required to supply an ORCID ID for their name upon submission of a revised manuscript (<https://orcid.org/>). Please find instructions on how to link your ORCID ID to your account in our manuscript

tracking system in our Author guidelines

<<https://www.embopress.org/page/journal/14693178/authorguide#authorshippinguidelines>>

8) We would also encourage you to include the source data for figure panels that show essential data. Numerical data should be provided as individual .xls or .csv files (including a tab describing the data). For blots or microscopy, uncropped images should be submitted (using a zip archive if multiple images need to be supplied for one panel). Additional information on source data and instruction on how to label the files are available at

<<https://www.embopress.org/page/journal/14693178/authorguide#sourcedata>>.

10) Regarding data quantification (see Figure Legends:

<https://www.embopress.org/page/journal/14693178/authorguide#figureformat>)

- the name of the statistical test used to generate error bars and P values,
- the number (n) of independent experiments (please specify technical or biological replicates) underlying each data point,
- the nature of the bars and error bars (s.d., s.e.m.),
- If the data are obtained from n {less than or equal to} 2, use scatter blots showing the individual data points.

Yours sincerely,

Referee #1:

The manuscript by Kaneko et al reports that ADAMTS2 is a critical regulator of radial migration in the developing neocortex, by activating the TGF β signaling pathway. In support of this assertion they showed that ADAMTS2 expression is induced during radial migration between E15 and E17 (Fig2A), especially in the IZ, SP and CP regions, and that ADAMTS2 knockdown (using siRNA) inhibits migration. Interestingly this phenotype can be reversed by co-transfecting siARN with a vector allowing the expression of siRNA-resistant ADAMTS2 mRNA.

I have a few questions and remarks intended to strengthen the manuscript.

The efficacy of the ADAMTS2-siRNA was tested in vitro in a completely different model. In order to link the brain phenotype with ADAMTS2 silencing, the efficacy of this siRNA should be also determined in the embryonic brain model.

I'm not a specialist of brain development, so perhaps my question is not appropriate, but what are the types of migrating cells? Only neurons or also glial GFAP-positive cells, because these cells are "more expected" to produce ADAMTS2. As usually observed with ISH, the resolution is far from optimal but would it be possible to determine an approximate number of positive cells?

Based on the rescue experiment (fig 3D) it's a bit surprising to see an inhibition of migration in the case of overexpression of ADAMTS2 (Fig 3C), because the exact dosage of any recombinant protein expression is usually hard to obtain. Were the experimental procedures identical for Fig3B and 3C?

Did the authors verify that siRNA transfection in Fig3B was not altered in the "rescue samples", so that their efficacy was similar in all the conditions? Again, it would have been most helpful to have qRT-PCR for evaluating the level of ADAMTS2 mRNA in the brain tissues.

If not all the cells express ADAMTS2 (see my remark above), there is a possibility that knockdown with siRNA (which can affect only cells expressing ADAMTS2) and overexpression by transfecting expression vector (causing expression in any transfected cells, even those which do not express endogenous ADAMTS2), do not necessarily reflect the same mechanism.

Page 6, line 98. It is wrong to say that the TGF β pathway is the main target of ADAMTS2. Procollagen processing is at least as critical.

Page 7, line 121. This is an oversimplification to state that the Chase ABC treatment mimics ECM remodeling. Please rephrase.

Page 9, Fig S3A. In the file that I received, it is impossible to read the table.

Figure 2. Why only 4 additional metalloproteinases are represented, while several other ADAMTS and MMP could potentially cleave proteoglycan and gelatin?

Page 11. It's highly surprising (so potentially highly interesting) to see a phenotype with samples from heterozygous animals and not from KO. And this is especially true with enzymes since, most of the time, 50% of their production is sufficient for the complete accomplishment of their functions. Compensation in KO by increased expression of another enzyme displaying the same activity is a potential explanation (although mainly identified in the zebrafish, to my best knowledge). Anyway it would have been most interesting to identify which metalloproteinase is responsible for this compensation by comparing their expression in heterozygous, KO and WT mice. Indeed, such information could participate to better define the mechanisms regulating neuron migration

Page 18, lines 319-321. What is the reason to mention the Loyes-Dietz syndrome?

Page 19, lines 329-330. Triple KO mice are impossible to produce since single ADAMTS3-KO die in utero around E14

Page 19, line 324. What do you mean by "reduced number of female pups" since the ratio of homozygous ADAMTS2-KO male and female is normal at birth? The childhood stroke has been shown in Human, and not in mice as the sentence would suggest.

Page 19, line 340. It is known that ADAMTS2 and ADAMTS3 can cleave reelin. Why the authors did not study the

consequences on TGF β signaling of Reelin cleavage by ADAMTS2?

Fig S4 D. It is written that Adamts2-KO mice showed impaired migration, while it's only seen for heterozygous.

Fig S6: It appears a little bit strange to see together "ECM remodeling" and "F-actin" which is part of the cytoskeleton. Perhaps put "actin" with "MP_BP transition"?

Referee #2:

In the current manuscript, Kaneko et al investigate the role of extracellular matrix in the subplate layer on radial migration of projection neurons. They identify ADAMTS2 and downstream TGF- β signalling as regulators of migration. Although authors brought pieces of evidence, the methods used are not always appropriate to address properly migration. As such, conclusions are often overstated. In addition the quality of the presented data (images, lack of quantification, missing labelling, poorly-described legend) is insufficient and figures are not at all self-sufficient (missing labelling, poorly-described legend). Overall, this manuscript does not meet the criteria for publication in Embo report. I would strongly recommend the authors to strengthen their data and increase the quality of the figures. Please find below list of suggestions:

Major comments:

1/ Authors claim that expression of ADAMTS2 is transient in migrating neurons. No data is supporting this statement.

2/ throughout the manuscript the authors used GFP expressed under the promoter CAGs, they should have used a neuron-specific promoter (DCX or NeuroD) to ensure that the phenotype they observed is a real migration phenotype and not a phenotype occurring earlier in progenitor (ADAMTS2 RNA being enriched in VZ)

On the same line: Authors used in utero electroporation of siRNA to knock-down Adamst2. Although its knock down clearly impairs neuronal migration, one cannot conclude if the impaired neuronal migration is solely due to defects in MP-BP transition of migrating neurons or defects in RG integrity or generation of neurons also contribute to those defects. To address this, as mentioned by the authors in the discussion, they should do a neuron specific knock-down of Adamst2 using shRNAs under NeuroD promoter or analyze the RG integrity and neurogenesis to figure out how/if defects in those contribute to the observed migration phenotype.

3/ Authors show an impaired migration upon ADAMTS2 KD (siRNA) that is rescued by Overexpression of ADAMTS2, how this result is compatible with the fact the Overexpression is leading itself to impair migration!!

4/ ADAMTS2 het cortices display a phenotype, whereas KO does not. Author suggest compensation. They could have tested this hypothesis by in utero electroporation (test ability of other ADAMTS to rescue ADAMTS2 KD phenotype). In addition, analysis of levels of ADAMTS proteins in WT versus het, versus HOM cortices would be very valuable.

5/ LTBP and Fibrillin2 are expressed in the SP, VZ and marginal zone (MZ) (Fig S1), TGFBR2 is expressed in all cortical layers from SP to MZ (Fig4) and Adamst2 is expressed in all layers apart from what is labeled as SVZ and SP (Fig2). However, scheme in Fig1A does not reflect the experimental data and are very misleading.

6/ SP enriched localization of Neurocan (Oohira et al, 1994) and Versican (Poop et al 2003) was already shown in developing rat brains. Different CPSGs were already shown to be involved in radial migration. Therefore, Fig1B and 1C does not bring anything new from my point of view so they could simply cite previous papers.

7/ As the main claim of the paper is defects in MP-BP transition, they should quantify parameters related to neuronal migration including number of neurons undergoing transition, time spent to transit from multipolar to bipolar, instead of only showing videos (Video S1/S2/S3). Along the same line, if they want to claim changed actin dynamics, they should quantify it as it is not clear from their videos (Video S2/S3).

8/ In Fig4J, authors show that CTGF, a downstream target of TGF β signaling, is significantly downregulated in Adamst2 HOM KO mice (that does not have a migration phenotype but not in HET mice that has the migration phenotype). As such, TGF- β signaling being the mediator of migration defects upon Adamst2 KD remains questionable. To solve this, authors could try to rescue the migration defects of ADAMTS2 KD or HET mice by re-introduction of TGF β (as they suggest this activates TGF- β signaling). Additionally, It would be very informative to do a RNA seq in GFP+ /Adamst2 siRNA + cells as it will allow analyzing the cell types and downstream pathways affected upon KD of Adamst2(to confirm a specific deregulation of TGF- β signaling).

9/ For Fig6, as it is a key experiment of the paper, it should be clearly indicated if the luminescent construct was co-electroporated with an RFP plasmid to use the RFP as an internal normalizer of electroporation efficiency. I guess the decrease in luciferase could also result from a weaker electroporation.

10/ ChaseABC treatment : validation of the efficiency of the treatment is missing - staining of ECM proteins would have helped.

Minor comments:

1) In Fig S1B, they should indicate that the GFP is coming from the Lpar1 -GFP mice they use to label the SP layer.

2) In fig 3B the effect of Adamst2 KD seem much weaker than the Fig3A (despite the very small standard error bar they had). Maybe they can select a better picture.

3) In FigS3A, it is not possible to read any of the gene names so that should be provided as an excel file

Referee #3:

The manuscript by Kaneko et al. examines the function of ADAMTS2 in the radial migration of pyramidal neurons in the mouse neocortex. The authors present data that ADAMTS2 is expressed during a key transition in the migration potential of pyramidal neurons. Gain or loss of function of ADAMTS2 perturbs migration of these cells to the cortical plate. The authors then infer from other published works that ADAMTS2 could regulate the availability of TGF signaling molecules they posit to regulate the multipolar transition of pyramidal cells as they encounter the SP. While I think aspects of the study are well done, and I agree with many of the conclusions the authors make, this study needs additional key data to support the overall conclusion of the paper. Fundamentally, their argument that ADAMTS2 regulates radial migration by activating TGF- β signaling is a correlation without further work. Yes, deleting ADAMTS2 reduces the TGF bioluminescence signal in the neocortex of reporter mice, but the group has not demonstrated a causal relationship between that phenomenon and pyramidal neuron migration using epistasis analysis. If their model of precisely controlled TGF signaling by ADAMTS2 is correct, then elevating TGF signaling in ADAMTS2 deficient neurons should rescue the migration block, and TGF loss of function should rescue the migration block due to ADAMTS2 gain of function. Such experiments, or additional epistasis studies the authors could devise to mitigate mere correlation, are required to transition the author's primarily correlative arguments into causative relationships.

Response to Reviewers

We thank the Referees for their valuable comments, which have improved our manuscripts.

We carefully considered all comments.

Referee #1:

The manuscript by Kaneko et al reports that ADAMTS2 is a critical regulator of radial migration in the developing neocortex, by activating the TGF β signaling pathway. In support of this assertion they showed that ADAMTS2 expression is induced during radial migration between E15 and E17 (Fig2A), especially in the IZ, SP and CP regions, and that ADAMTS2 knockdown (using siRNA) inhibits migration. Interestingly this phenotype can be reversed by co-transfecting siARN with a vector allowing the expression of siRNA-resistant ADAMTS2 mRNA.

I have a few questions and remarks intended to strengthen the manuscript.

The efficacy of the ADAMTS2-siRNA was tested *in vitro* in a completely different model. In order to link the brain phenotype with ADAMTS2 silencing, the efficacy of this siRNA should be also determined in the embryonic brain model.

Thank you for pointing this out. We introduced si-RNA along with GFP-expression plasmids into neural progenitor cells by *in utero* electroporation. Adamts2 knockdown cells were isolated from the electroporated cortex by FACS according to GFP-positivity, and Q-PCR was performed. The results showed that *Adamts2* mRNA was reduced to approximately 60% of the control cells (Fig. S4B). Knockdown efficiency was low compared with that from NIH3T3 cells, probably because the purity of the FACS-sorted GFP-positive neurons was low. Knockdown of Adamts2 was also confirmed by double staining of *in situ* hybridization and GFP immunostaining (Fig. S4C).

We have added Figs. S4B,C and text (lines 189-195).

I'm not a specialist of brain development, so perhaps my question is not appropriate, but what are the types of migrating cells? Only neurons or also glial GFAP-positive cells, because these cells are "more expected" to produce ADAMTS2. As usually observed with ISH, the resolution is far from optimal but would it be possible to determine an approximate

number of positive cells?

In this study, we labeled neural progenitor cells localized on the ventricular surface by *in utero* electroporation at E14. At this stage, only neurons were labeled. Gliogenesis begins around E17 after the end of neurogenesis. Thus, we only observed neurons because glial cells have not yet emerged.

Based on the rescue experiment (fig 3D) it's a bit surprising to see an inhibition of migration in the case of overexpression of ADAMTS2 (Fig 3C), because the exact dosage of any recombinant protein expression is usually hard to obtain. Were the experimental procedures identical for Fig3B and 3C?

Because overexpression can inhibit migration, as shown in Fig. 3B, we carefully determined the concentration for the rescue experiments. We first tried three different concentrations (0.5, 1.0, and 2.0 $\mu\text{g}/\mu\text{l}$) of si-RNA-resistant *Adamts2* expression plasmid. The results showed that 0.5 $\mu\text{g}/\mu\text{l}$ was the most efficient for rescue, while higher concentrations were not effective (Fig. S6). In the case of 2.0 $\mu\text{g}/\mu\text{l}$, migration was rather suppressed (Fig. S6). This is consistent with the results of overexpression-induced migration impairment (Fig. 3B). Therefore, too much or too little ADAMTS2 leads to impaired migration. In addition, *in utero* electroporation was always performed under the same conditions.

We have added Fig. S6 and text (lines 206-212).

Did the authors verify that siRNA transfection in Fig3B was not altered in the "rescue samples", so that their efficacy was similar in all the conditions? Again, it would have been most helpful to have qRT-PCR for evaluating the level of ADAMTS2 mRNA in the brain tissues.

Very high doses of expression plasmid may suppress si-RNA incorporation. However, in our experiments, rescue only occurred with a low dose of expression plasmid (0.5 $\mu\text{g}/\mu\text{l}$) for 10 $\mu\text{g}/\mu\text{l}$ of si-RNA. Even under this knockdown condition, a high dose of *Adamts2*-expression plasmid (2.0 $\mu\text{g}/\mu\text{l}$) resulted in impaired migration (Fig. S6). This indicated that appropriate levels of *Adamts2* are crucial.

If not all the cells express ADAMTS2 (see my remark above), there is a possibility that

knockdown with siRNA (which can affect only cells expressing ADAMTS2) and overexpression by transfecting expression vector (causing expression in any transfected cells, even those which do not express endogenous ADAMTS2), do not necessarily reflect the same mechanism.

As mentioned above, both si-RNA and expression plasmids are introduced into a relatively homogeneous population of neural progenitors that produce only neurons by *in utero* electroporation. It is also known that neural progenitors uptake both the co-electroporated substances, suggesting that both si-RNA and expression plasmids were introduced into the same neurons. The expression of endogenous ADAMTS2 in these neurons is up-regulated during migration, especially around the SP layer (Fig. 2A). Moreover, the expression of plasmid-derived ADAMTS2 is not strictly regulated. However, appropriate levels of ADAMTS2 overexpression may rescue the migration defects of knockdown neurons. High doses of ADAMTS2 expression plasmid may result in the overexpression of this proteinase at inappropriate amounts, positions, and times, leading to migration defects.

Page 6, line 98. It is wrong to say that the TGFb pathway is the main target of ADAMTS2. Procollagen processing is at least as critical.

Thank you for pointing this out. We have amended the wording and rewritten it as "one of the targets" (line 105).

Page 7, line 121. This is an oversimplification to state that the Chase ABC treatment mimics ECM remodeling. Please rephrase.

We have changed that part as follows;

"CHase ABC is a bacterial enzyme that degrades chondroitin sulfate and hyaluronan, and thus, breaks down ECM structures and may affect the ECM-dependent signaling processes." (line 134-136).

Page 9, Fig S3A. In the file that I received, it is impossible to read the table. Sorry for the inconvenience. We attached an Excel file as Table 1.

Figure 2. Why only 4 additional metalloproteinases are represented, while several other ADAMTS and MMP could potentially cleave proteoglycan and gelatin?

We have microarray data for different stages of migrating neurons used in our previous study (Ohtaka-Maruyama et al., Science, 2018). These data extracted 2,377 probes for genes with significantly variable expression as the neurons migrated. In the present study, we re-analyzed that data and extracted genes with ECM-related Gene Ontology terms. As a result, 195 genes were extracted, as listed in Table1. Among these genes, there were four ECM-degrading enzymes in addition to Adamts2, such as Adamts1, Adamts10, MMP 15, and MMP17, so we focused on these enzymes. We describe the details in the Results (lines 163-174). We also focused on Adamts3 and 14 as subfamily members of Adamts2 (Fig. S3). Among these, Adamts2 was remarkably up-regulated during migration, so we focused on this gene.

Page 11. It's highly surprising (so potentially highly interesting) to see a phenotype with samples from heterozygous animals and not from KO. And this is especially true with enzymes since, most of the time, 50% of their production is sufficient for the complete accomplishment of their functions. Compensation in KO by increased expression of another enzyme displaying the same activity is a potential explanation (although mainly identified in the zebrafish, to my best knowledge). Anyway it would have been most interesting to identify which metalloproteinase is responsible for this compensation by comparing their expression in heterozygous, KO and WT mice. Indeed, such information could participate to better define the mechanisms regulating neuron migration.

As discussed above, the expression of ADAMTS2 must be strictly regulated during neuronal migration. It is likely that such strict regulation is not possible under heterozygous conditions (50% production). However, genetic compensation can occur during development in the absence of the Adamts2 gene. We observed increased expressions of Adamts3 and Adamts14 in the cerebrum of homozygous Adamts2 KO mice, although they were not significant (lines 224-227, see attached figure). We also demonstrated that neuronal expression of Adamts3 can rescue the knockdown phenotype of Adamts2 (Fig. S7; lines 219-227). These findings suggest that metalloproteinases, such as Adamts3, rescued the migration defects in the homozygous mice. We would like to conduct more comprehensive analyses in the future.

Page 18, lines 319-321. What is the reason to mention the Loyes-Dietz syndrome?

We removed the example of LD syndrome because it is not directly related to the regulation of cell migration.

Page 19, lines 329-330. Triple KO mice are impossible to produce since single ADAMTS3-KO die in utero around E14

Thank you for pointing this out. We have removed this sentence.

Page 19, line 324. What do you mean by "reduced number of female pups" since the ratio of homozygous ADAMTS2-KO male and female is normal at birth? The childhood stroke has been shown in Human, and not in mice as the sentence would suggest.

Thank you for pointing this out. We have rewritten this part (lines 383-387).

Page 19, line 340. It is known that ADAMTS2 and ADAMTS3 can cleave reelin. Why the authors did not study the consequences on TGF β signaling of Reelin cleavage by ADAMTS2?

Adamts3 is expressed in the cortical plate of the embryonic brain and is responsible for the cleavage of Reelin secreted by Cajal-Retzius cells located in the marginal zone (MZ) (Ogino et al., 2017). On the contrary, Adamts2's Reelin cleavage activity has been detected in the cortex and hippocampus of the adult brain (Yamakage et al., 2019). In the present study, we found that Adamts2 is expressed in mid-embryonic migratory neurons, especially multipolar neurons around the SP layer. Thus, it is likely that TGF- β signaling activated by ADAMTS2 is separated from Reelin cleavage.

Fig S4 D. It is written that Adamts2-KO mice showed impaired migration, while it's only seen for heterozygous.

Thank you for pointing this out. The description needed to be corrected. We have rewritten it

(Fig. S7) as follows.

“Adamts2 KO mice showed significant migration defects in heterozygous (HT) mice, but not in homozygous (HM) mice.”

Fig S6: It appears a little bit strange to see together "ECM remodeling" and "F-actin" which is part of the cytoskeleton. Perhaps put "actin" with "MP_BP transition"?

Thank you for noticing. We have moved “F-actin” to the side of the MP-BP transition.

Referee #2:

In the current manuscript, Kaneko et al investigate the role of extracellular matrix in the subplate layer on radial migration of projection neurons. They identify ADAMTS2 and downstream TGF-beta signalling as regulators of migration. Although authors brought pieces of evidence, the methods used are not always appropriate to address properly migration. As such, conclusions are often overstated. In addition the quality of the presented data (images, lack of quantification, missing labelling, poorly-described legend) is insufficient and figures are not at all self-sufficient (missing labelling, poorly-described legend). Overall, this manuscript does not the criteria for publication in Embo report. I would strongly recommend the authors to strengthen their data and increase the quality of the figures. Please find below list of suggestions:

Major comments:

1/ Authors claim that expression of ADAMTS2 is transient in migrating neurons. No data is supporting this statement.

Our data in Fig. 2A demonstrated that Adamts2 expression was up-regulated from E15 to E17 in FACS-sorted neurons during the multipolar-to-bipolar transition. Furthermore, *in situ* hybridization results (Fig. 2B) showed a strong IZ signal at E17, but weak in the CP, which implied that the expression is transiently up-regulated in migrating neurons. However, it does not directly indicate that the expression was transient. Thus, we modified the wording and removed the word "transient".

2/ throughout the manuscript the authors used GFP expressed under the promotor CAGs, they should have used a neuron-specific promotor (DCX or NeuroD) to ensure that the phenotype they observed is a real migration phenotype and not a phenotype occurring earlier in progenitor (ADAMTS2 RNA being enriched in VZ)

On the same line: Authors used in utero electroporation of siRNA to knock-down Adamst2. Although its knock down clearly impairs neuronal migration, one cannot conclude if the impaired neuronal migration is solely due to defects in MP-BP transition of migrating neurons or defects in RG integrity or generation of neurons also contribute to those defects. To address this, as mentioned by the authors in the discussion, they should do a neuron specific knock-down of Adamst2 using shRNAs under NeuroD promoter or analyze the RG integrity and neurogenesis to figure out how/if defects in those contribute to the observed migration phenotype.

To determine whether Adamts2 overexpression and knockdown affect progenitor cells of the ventricular zone, we stained mitotic cells with Ki67 and intermediate progenitor cells with Tbr2 and calculated the percentage of GFP-positive cells. As shown in Fig. S5, there was no significant difference in the percentage between Control and Adamts2-overexpression, or knockdown brains. This suggests that, in the present study, the phenotype of impaired migration observed in Adamts2 overexpression and si-RNA-mediated knockdown was not caused by effects on progenitor cells such as abnormal cell cycle exit. We also performed experiments using a NeuroD promoter (see below).

3/ Authors show an impaired migration upon ADAMTS2 KD (siRNA) that is rescue by Overexpression of ADAMTS2, how this result is compatible with the fact the Overexpression is leading itself to impair migration!!

Because overexpression would inhibit migration as shown in Fig. 3B, we carefully determined the concentration for the rescue experiments. For rescue experiments, we first tried three different concentrations (0.5, 1.0, and 2.0 $\mu\text{g}/\mu\text{l}$) of si-RNA-resistant *Adamts2* expression plasmid. The results showed that 0.5 $\mu\text{g}/\mu\text{l}$ was the most efficient for rescue, whereas higher concentrations were not effective (Fig. S6). In the case of 2.0 $\mu\text{g}/\mu\text{l}$, migration was rather suppressed (Fig. S6). This is consistent with the results of

overexpression-induced migration impairment (Fig. 3B). Therefore, too much or too little ADAMTS2 leads to impaired migration. We have added Fig. S6 and text (lines 206-212).

4/ ADAMTS2 het cortices display a phenotype, whereas KO does not. Author suggest compensation. They could have tested this hypothesis by *in utero* electroporation (test ability of other ADAMTS to rescue ADAMTS2 KD phenotype). In addition, analysis of levels of ADAMTS proteins in WT versus het, versus HOM cortices would be very valuable.

We investigated whether other Adamts can rescue migration defects in Adamts2 knockdowns using *in utero* electroporation. (Fig. S8). We constructed new expression plasmids, in which Adamts3 and Adamts14 cDNAs, which belong to the same subfamily as Adamts2, were subcloned directly under the NeuroD1 promoter. The results showed that Adamts3 can rescue the knockdown phenotype of Adamts2. We also observed increased expressions of Adamts3 and Adamts14 in the cerebrum of homozygous Adamts2 KO mice, although they were not significant (lines 224-227, attached figure). These findings suggest that metalloproteinases, such as Adamts3, rescued the migration defects in the homozygous mice.

Regarding the protein levels, we could only perform minimal crossbreeding experiments to obtain samples of each genotype for RNA preparation during the paper's revision period, so we could not obtain the brains for protein preparation. In the future, we would like to analyze the levels of ADAMTS proteins.

5/ LTBP and Fibrillin2 are expressed in the SP, VZ and marginal zone (MZ) (Fig S1), TGFBR11 is expressed in all cortical layers from SP to MZ (Fig4) and Adamst2 is expressed in all layers apart from what is labeled as SVZ and SP (Fig2). However, scheme in Fig1A does not reflect the experimental data and are very misleading.

In Fig. 1A, we focused on the expression around the SP layer. However, we agree that it is misleading. Therefore, we have revised the figure. As for Adamts2, the ISH results show that it is not uniformly expressed in all layers. As shown in Fig. 2B, IZ is strongly stained in E16 and 17.

6/ SP enriched localization of Neurocan (Oohira et al, 1994) and Versican (Poop et al 2003)

was already shown in developing rat brains. Different CPSGs were already shown to be involved in radial migration. Therefore, Fig1B and 1C does not bring anything new from my point of view so they could simply cite previous papers.

We have removed the immunostained photographs of neurocan from Fig. 1B and only mentioned them in the text. Regarding versican, we conducted additional experiments to confirm which isoforms of versican are expressed in the cortex. Furthermore, we added new data demonstrating that cleaved versican is localized around SpNs. As for Fig. 1C, we believe that the detailed behaviors of migrating neurons in the presence of CHase ABC are novel. Therefore, we did not remove Fig. 1C.

7/ As the main claim of the paper is defects in MP-BP transition, they should quantify parameters related to neuronal migration including number of neurons undergoing transition, time spent to transit from multipolar to bipolar, instead of only showing videos (Video S1/S2/S3). Along the same line, if they want to claim changed actin dynamics, they should quantify it as it is not clear from their videos (Video S2/S3).

We quantified the video data: for Adamts2 knockdown, we measured the speed of migration in the direction of the brain surface (Top panel of Fig. S9C). The results showed that the migration speed was significantly reduced in the Adamts2 knockdown neurons. Although counting the number of neurites is an appropriate way to quantify actin dynamics, we could not count them in individual cells because of the high density of cells. Instead, we counted the number of multipolar and bipolar cells. In the control sections, bipolar cells increased during 10 h of culture. However, in the Adamts2 knockdown sections, little increase was observed. Thus, the claim in the video that the multipolar-to-bipolar (MP-BP) transition process is impaired by Adamts2 knockdown is further supported (Fig. S9C, lower panels). For TGF- β inhibitors, we also quantified the MP:BP ratio from the video data and found that the MP-BP transition was impaired in the inhibitor-treated group (Fig. 5C).

8/ In Fig4J, authors show that CTGF, a downstream target of TGFB signaling, is significantly downregulated in Adamst2 HOM KO mice (that does not have a migration phenotype but not in HET mice that has the migration phenotype). As such, TGF-B signaling being the mediator of migration defects upon Adamst2 KD remains questionable. To solve this,

authors could try to rescue the migration defects of ADAMST2 KD or HET mice by re-introduction of TGF β 1 (as they suggest this activates TGF- β signaling). Additionally, It would be very informative to do a RNA seq in GFP+ /Adamst2 siRNA + cells as it will allow analyzing the cell types and downstream pathways affected upon KD of Adamst2(to confirm a specific deregulation of TGF- β signaling).

The expression of CTGF was measured using mRNA samples from the whole cerebrum. Therefore, the results suggest that whole TGF- β signaling was reduced in the whole cerebrum of homozygous Adamts2 KO mice. Furthermore, the migration phenotype reflects the local TGF- β signaling in the specific neurons at the specific developmental stage. To increase local TGF- β signaling in the migrating neurons of Adamts2KO heterozygous mice, TGF- β RII and TGF- β 2 were overexpressed in these neurons using NeuroD1-promoter. As a result, the migration defect was rescued by overexpression of TGF- β 2 (Fig. 6G), suggesting that the TGF- β signaling is downstream of Adamts2. We couldn't conduct RNA sequencing analysis because of insufficient time. We would like to consider this in the future.

9/ For Fig6, as it is a key experiment of the paper, it should be clearly indicated if the luminescent construct was co-electroporated with an RFP plasmid to use the RFP as an internal normalizer of electroporation efficiency. I guess the decrease in luciferase could also result from a weaker electroporation.

As an internal control to normalize electroporation efficiency, we used a CAG-GFP plasmid which was introduced together with a luciferase reporter. The GFP signal was still visible, during the locomotion in the CP even after the luminescence signal had weakened. We calculated the mean fluorescence values of the GFP signal. They are 3017.9 for Control, 3021.9 for inhibitor treated and 3247.3 for si-Adamts2. So, there is no extreme difference in electroporation efficiency. In our revised version, we also displayed images of the GFP signal corresponding to the respective luminescence imaging (Fig.6B-D).

10/ ChaseABC treatment: validation of the efficiency of the treatment is missing - staining of ECM proteins would have helped.

The efficiency of CHase ABC treatment was confirmed by immunohistochemical staining

with anti-CS monoclonal antibody (Fig. S1C).

Minor comments:

1) In Fig S1B, they should indicate that the GFP is coming from the Lpar1 -GFP mice they use to label the SP layer.

We revised the labels of Fig. S1 with Lpar1-EGFP, as suggested.

2) In fig 3B the effect of Adamst2 KD seem much weaker than the Fig3A (despite the very small standard error bar they had). Maybe they can select a better picture.

Even within the same group, there is variation in the distribution of migrating neurons between individual samples. We used another representative image of the group quantified in this study (Fig. 3C).

3) In FigS3A, it is not possible to read any of the gene names so that should be provided as an excel file

Sorry for the inconvenience. We attached an Excel file for this list as Table 1.

Referee #3:

The manuscript by Kaneko et al. examines the function of ADAMTS2 in the radial migration of pyramidal neurons in the mouse neocortex. The authors present data that ADAMTS2 is expressed during a key transition in the migration potential of pyramidal neurons. Gain or loss of function of ADAMTS2 perturbs migration of these cells to the cortical plate. The authors then infer from other published works that ADAMTS2 could regulate the availability of TGF signaling molecules they posit to regulate the multipolar transition of pyramidal cells as they encounter the SP. While I think aspects of the study are well done, and I agree with many of the conclusions the authors make, this study needs additional key data to support the overall conclusion of the paper. Fundamentally, their argument that ADAMTS2 regulates radial migration by activating TGF- β signaling is a correlation without further work. Yes, deleting ADAMTS2 reduces the TGF bioluminescence signal in the neocortex of reporter mice, but the group has not demonstrated a causal relationship between that phenomenon and pyramidal neuron migration using epistasis analysis. If their model of precisely

controlled TGF signaling by ADAMTS2 is correct, then elevating TGF signaling in ADAMTS2 deficient neurons should rescue the migration block, and TGF loss of function should rescue the migration block due to ADAMTS2 gain of function. Such experiments, or additional epistasis studies the authors could devise to mitigate mere correlation, are required to transition the author's primarily correlative arguments into causative relationships.

If TGF- β signaling is downstream of Adamts2 in the migrating neurons, raising TGF- β signaling in these neurons will rescue the phenotype of the Adamts2 KO heterozygous cortex. Therefore, we tested whether overexpression of TGF- β RII or TGF- β 2 can rescue this phenotype in the heterozygous mice using the NeuroD1 promoter. The results showed that overexpression of TGF- β 2 rescued the migration defect, as shown in Fig. 6H. This result strongly suggests the presence of TGF- β signaling downstream of Adamts2.

Dear Chiaki,

Thank you for the submission of your revised manuscript. We have now received the full set of referee reports as well as referee cross-comments that are all pasted below.

As you will see, the referees acknowledge that the revised ms has been improved. However, they also still have some concerns that should be addressed, including more experimentation. From the cross-comments it becomes clear that the second rescue experiment (overexpressing Adamts2 and knocking down TGFbeta) is not strictly required. However, if possible, the addition of this experiment would certainly be very welcome. If this experiment will not be performed, please follow referee 3's suggestions below. All other final concerns need to be fully addressed, and the quality of the figures and the images needs to be improved. Please co-submit a point-by-point response to all final comments with your newly revised ms.

A few editorial requests will also need to be addressed:

- Please add up to 5 keywords to the ms file.
- A "Data Availability Section" needs to be added to the end of the materials and methods. This section should list all links to data generated in this study and deposited in public databases.
- A "Disclosure and Competing Interest Statement" must be added.
- Please correct the reference format to the EMBO reports (Harvard) style.
- "Data not shown" on page 10 must be removed, as per journal policy. Please either show the data or rewrite.
- The author checklist has not been fully completed. Please send us a fully completed checklist with your newly revised ms.
- Some funding information is missing in our online submission system, please add all info when you upload the newly revised ms.
- All main and EV figures need to be uploaded as individual files.
- Each movie should be zipped together with its legend and uploaded as one individual file.
- The Significance statement should be removed.
- The manuscript sections are in the wrong order. The order of the ms sections should be: abstract, introduction, results, discussion, materials & methods, data availability section, acknowledgments, disclosure statement and competing interests, references, main figure legends, EV figure legends. Please correct.
- Please add the heading 'Figure Legends'.
- The main figures must fit onto one page, Fig 6 runs over 2 pages, please correct.
- There are 11 supplementary figures. Some run over 2 pages so are not a suitable format for an EV figure. You can choose 5 supplementary figures that can be uploaded as EV figures. EV figures are integrated into the ms text online and expand when clicked. The remaining figures can be combined in the Appendix file. Please see our guide to authors for more information. Please also correct all figure callouts in the ms text.
- I attach to this email a related ms file with comments by our data editors. Please address all comments in the final ms.

I would like to suggest some changes to the title and abstract. Please let me know whether you agree with the following:

ADAMTS2 promotes radial migration by activating TGF- β signaling in the developing neocortex

The mammalian neocortex is formed by sequential radial migration of newborn excitatory neurons. Migrating neurons undergo a multipolar-to-bipolar transition at the subplate (SP) layer, where extracellular matrix (ECM) components are abundantly expressed. Here, we investigate the role of the ECM at the SP layer. We show that TGF- β signaling-related ECM proteins, and their downstream effector, p-smad2/3, are selectively expressed in the SP layer. We also find that migrating neurons express a disintegrin and metalloproteinase with thrombospondin motif 2 (ADAMTS2), an ECM metalloproteinase, just below the SP layer. Knockdown and knockout of Adamts2 suppresses the multipolar-to-bipolar transition of migrating neurons and disturbs radial migration. Time-lapse luminescence imaging of TGF- β signaling indicates that ADAMTS2 activates this signaling pathway in

migrating neurons during the multipolar to bipolar transition at the SP layer. Overexpression of TGF- β 2 in migrating neurons partially rescues migration defects in ADAMTS2 knockout mice. Our data suggest that ADAMTS2 secreted by the migrating multipolar neurons activates TGF- β signaling by ECM remodeling of the SP layer, which might drive the multipolar to bipolar transition.

EMBO press papers are accompanied online by A) a short (1-2 sentences) summary of the findings and their significance, B) 2-3 bullet points highlighting key results and C) a synopsis image that is exactly 550 pixels wide and 200-600 pixels high (the height is variable). You can either show a model or key data in the synopsis image. Please note that text needs to be readable at the final size. Please send us this information along with the final manuscript.

I look forward to seeing a newly revised form of your manuscript when it is ready.

Referee #1:

I'm satisfied by the responses.

Just two mistakes in the legend of fig S4:

"... and GFP-positive cells were isolated by FACS and Q-PCR". Should we understand "and Q-PCR were performed"?

"Adamts2" and not "Admts2"

Referee #2:

The authors made subsequent work to improve the manuscript. I acknowledge the attempts to answer to the comments. However, I noticed that some quantifications are still missing and that the legends/description of the figure could have been improved by adding more details.

The authors have removed the claiming about transient expression of ADAMTS2. Still their conclusions are based on transcripts expression. Analysis at the protein level would have been much more informative.

To address the point on non-cell autonomous effect on progenitors, authors have analysed number of Tbr2 + intermediate progenitors. It would again have been more informative to analyse the number of Apical progenitors whose basal process serves as guide for neuronal migration (Pax6 staining) and add a nestin staining to address the integrity of the radial scaffold (that could be impaired even if proliferation and number of progenitors are not). Authors tried to answer to that point showing no changes of proliferation, however the quality of the ki67 staining presented in the figure make me doubt about the feasibility and reliability of these countings (we have hard time to see the expected nuclear staining). I suggested to perform the knock-down and overexpression of ADAMTS2 using a neuro D promotor instead of a PCAGGS promotor to address its function in neurons exclusively. Although, authors have done some rescue with neuro-D adamts3 and Adamts14 that attest of a neuron-specific phenotype (but not complete rescue according to the numbers in figure S8- stats are missing to make sure about a complete rescue (bin-bin comparison)), they do not have tested the effect of ADAMts2 per se in neuron only. I think this is a key experiment (neuroD-driven expression or rescue with ND-ADAMTS2).

About figure 6 and normalizing the luminescent signal on the IUE: The images provided are of very bad quality, we barely see individual cells. I am not convinced that authors, by quantifying such images could obtain reliable quantification. In addition in figure 6E and F, they follow luminescent signal in migrating neurons and claim for a drop after conversion, as we cannot see the polarity (MP versus BP) in those images, I don't get how they came to that conclusion. Saying that there is a drastic decrease as migration proceed would be more accurate.

I think that including quantification of the time lapse improve greatly the quality of the manuscript. I would have liked to see the quantification of the velocity and /or MP-BP conversion in CHASE ABC experiments (1C).

Other points:

Figure 1D: what is RFP ? an electroporated plasmid? At which stages, information are missing from the legend.

In situ zymography images (control) are more convincing in the S2 than in main figure.

Table : E15 (expression) column, I guess this a fold change as negative value are shown.

Figure S4C: images not convincing

Line 230: authors should mention that rescue with neuroD-ADATS3 show that at least some of the observed phenotype is neuron-specific.

F-actin related data: authors are providing movies, Quantification included as figure in the paper would help the reading and the understanding of the actin-related phenotype? :line 237 authors mentioned F-actin dynamics, which parameters have been analysed?

4I: why not doing the TGF- β RII staining in Lpar1-eGFP to ensure the localization at SP ?

Referee #3:

I didn't ask for extensive revisions in my previous review but requested two critical data to demonstrate causality in the author's system. The epistasis tests I suggested were required to fully elucidate the role of TGF-Beta signaling in relation to ADAMTS2. The authors provided what happened when TGF-Beta gain of function was made in the ADAMTS2 loss of function context. Curiously, the authors are conspicuously silent on the second half of the epistasis test, namely, if TGF loss of function should rescue the migration block due to ADAMTS2 gain of function. I would still require this experiment to buy into the author's proposed model fully.

Cross-comments by referee 1:

Just to be clear, I was satisfied with the authors' responses regarding my personal remarks which were more focused on the ADAMTS enzymes.

I am not a specialist in neurology, but it seems that the remarks of the other two reviewers are relevant and worth addressing.

Cross-comments by referee 2:

I saw that authors are willing to answer my concerns. I think this will definitively strengthen their conclusion. I would also encourage them to improve the overall quality of the figures and images provided.

Concerning the relationship between TGF- β and ADAMTS2, I also asked for the rescue of the ADAMTS2 KD phenotype by increase of TGF- β signalling in the first round of revision. Although the reverse experiment could be interesting, I am not sure it would bring much to the conclusion. However, you must note that Conny Cepko has developed strategy to express miRNA under any kind of polIII promoter (PCAGGS miR30 plasmids can be found in addgene). Authors will have to replace the pCAGGS promoter by the NeuroD and insert the miRNAs; it would require maximum a month or so of cloning and then 2-3 month to complete the in utero electroporation experiments.

Cross-comments by referee 3:

The NeuroD promoter experiment seems quite feasible, and it is an important part of the author's argument. In terms of my rationale for requesting the two experiments in my original review and requesting the TGF β loss of function in the second review: given the correlative nature of a good number of the original experiments, I asked for these epistasis experiments to get at causality in their system. I re-asked for the TGF β loss of function experiment because the authors didn't even acknowledge that specific experiment negatively or positively in their rebuttal. I potentially understand their predicament: I've had a similar situation in a recent paper submission where silencing a gene on a single-cell basis did not yield a phenotype but knocking the same gene out in a whole cell population did (a non-cell autonomous function). If they can't do my experiment without an extensive delay, I'd like to see two things 1) they should change the language of their discussion to reflect the lack of full causality in their working model, and 2) if they did the TGF β silencing but saw a negative result they should report it perhaps in the form of a supplemental figure with appropriate discussion as it relates to their working model.

Response to Reviewers

We thank the referees for their valuable comments and helping to improve our manuscripts. We carefully considered all comments.

Referee #1:

I'm satisfied by the responses.

Just two mistakes in the legend of fig S4:

"... and GFP-positive cells were isolated by FACS and Q-PCR". Should we understand "and Q-PCR were performed"?

"Adamts2" and not "Admts2"

Thank you for pointing these out. We revised the legend of FigS4 accordingly.

Referee #2:

The authors made subsequent work to improve the manuscript. I acknowledge the attempts to answer to the comments. However, I noticed that some quantifications are still missing and that the legends/description of the figure could have been improved by adding more details.

Thank you for your comment. In response to your comments, we have made several improvements.

The authors have removed the claiming about transient expression of ADAMTS2. Still their conclusions are based on transcripts expression. Analysis at the protein level would have been much more informative.

As you say, examining the expression status at the protein level is important. However, Adamts2 antibodies are difficult to stain in tissue sections, unfortunately, so we analyzed them at the RNA expression level.

To address the point on non-cell autonomous effect on progenitors, authors have analysed number of Tbr2 + intermediate progenitors. It would again have been more informative to analyse the number of Apical progenitors whose basal process serves

as guide for neuronal migration (Pax6 staining) and add a nestin staining to address the integrity of the radial scaffold (that could be impaired even if proliferation and number of progenitors are not). Authors tried to answer to that point showing no changes of proliferation, however the quality of the ki67 staining presented in the figure make me doubt about the feasibility and reliability of these countings (we have hard time to see the expected nuclear staining).

Thank you for pointing this out. We have now added nestin and Pax6 staining to our analysis, and we found no change in the appearance of radial glial fibers as visualized by nestin in the electroporated and non-electroporated cortical regions of Adamts2 knockdown or overexpressed cortices (Fig.S6F). There was also no significant difference in the number of Pax6 positive cells (Fig.S6F right). For Ki67 staining, we provided an enlarged image for one example, showing that we could adequately count the overlap with GFP-positive cells (see attached figure for reviewers). We have also attached an image of the magnification used for the actual count (Fig.S6, C-E).

I suggested to perform the knock-down and overexpression of ADAMTS2 using a neuro D promotor instead of a PCAGGS promotor to address its function in neurons exclusively. Although, authors have done some rescue with neuro-D adamts3 and Adamts14 that attest of a neuron-specific phenotype (but not complete rescue according to the numbers in figure S8- stats are missing to make sure about a complete rescue (bin-bin comparison)), they do not have tested the effect of ADAMts2 per se in neuron only. I think this is a key experiment (neuroD-driven expression or rescue with ND-ADAMTS2).

We created a pNeuroD1-Adamts2 plasmid and performed overexpression experiments (Fig.S5). The results showed that Adamts2 overexpression of Adamts2 under the Neuro D1 promoter also resulted in delayed neuronal migration, similar to the CAG promoter. Therefore, we found that the proper amount of Adamts2 expression in neurons is essential. For the knockdown experiments, neuron-specific promoters could not be used because the si-RNA was directly introduced by in utero electroporation. Considering the Adamts3 rescue experiment, and there was a significant difference between the Adamts2 KD and Adamts3 rescue groups. Therefore, Adamts3 overexpression rescues Adamts2KD.

About figure 6 and normalizing the luminescent signal on the IUE: The images provided are of very bad quality, we barely see individual cells. I am not convinced that authors, by quantifying such images could obtain reliable quantification.

As you mentioned, the image quality is compromised. However, this is due to the resolution limitations of the luminescence imaging equipment we utilized, the ATTO Cellgraph, a widely recognized tool in the field of luminescent imaging.

Therefore, as meticulously detailed in the Materials and Methods section, we undertook a comprehensive approach to quantify the degree of luminescence and its temporal changes from these images, described as follows.

“The data analysis was performed using Python 3.8.0. The luminescence image was analyzed in a grid of 16 x 14 squares, and the average luminance value of each square was calculated. The average of the signals along the time axis of each grid was Fourier transformed. We used these values to visualize changes in luminance over time. A low-pass filter was applied by adding frequencies of 30 to 75 Hz for each grid, to remove high outliers caused by cosmic rays. The location of the SP layer was then confirmed in the brightfield image, and luminance values were obtained for only the top and bottom four rows of the SP layer. Because of the high background obtained immediately after the start, the first 4.5 h were excluded from the analysis. The frequency domain was then summed from 2 to 30 Hz for each grid, and the results were represented in a violin plot.” (Lines 766–776)

In addition, in figure 6E and F, they follow luminescent signal in migrating neurons and claim for a drop after conversion, as we cannot see the polarity (MP versus BP) in those images, I don't get how they came to that conclusion. Saying that there is a drastic decrease as migration proceed would be more accurate.

Indeed, the cell morphology is not clearly visible, so we have changed the expression as follows, according to your suggestion.

“The magnified time-lapse images reveals that the MpNs in the lower part of the SP layer temporarily showed a strong luminescence signal, but then the signal vanished (Fig.7E). These results suggest that ADAMTS2 activates TGF- β signaling in the MpNs around the bottom of the SP layer, drastically reducing it after the transition

from multipolar to bipolar.” (Lines 305–309)

I think that including quantification of the time lapse improve greatly the quality of the manuscript. I would have liked to see the quantification of the velocity and /or MP–BP conversion in CHASE ABC experiments (1C).

As suggested, we quantified and graphed the migration speed of the CHaseABC–treated cells using time–lapse imaging data (Fig.1D). We confirmed that the speed was significantly lower in the CHaseABC–treated group compared to the control.

Other points:

Figure 1D: what is RFP ? an electroporated plasmid? At which stages, information are missing from the legend.

We apologize for the lack of clarity. RFP is a plasmid introduced by in utero electroporation to label migrating neurons; it was mentioned in the Methods section, but we also added this to the figure legend (Fig.S2, previously Fig.1D).

In situ zymography images (control) are more convincing in the S2 than in main figure.

Following your suggestion, we swapped Figs.1E and FigS2.

Table : E15 (expression) column, I guess this a fold change as negative value are shown.

For E15, the expression is shown as \log_2 , so if the expression is less than 1, it is negative; for E16 and 17, the fold change is also shown as \log_2 .

Figure S4C: images not convincing

In situ hybridization and antibody double staining are commonly used methods based on a color reaction. Purple and brown colors are attributed to positive cells for both mRNA and antibody staining signals, but these signals are generally somewhat challenging to see. In a previous paper (<https://onlinelibrary.wiley.com/doi/10.1002/cne.21350>), we used the same method

to identify double-positive cells for transcription factors (Fig. 4 of the attached paper; we have uploaded this paper for your reference.) The present data also shows that the purple *Adamts2* mRNA signal is reduced in GFP-positive knockdown cells.

Line 230: authors should mention that rescue with *neuroD-ADATS3* show that at least some of the observed phenotype is neuron-specific.

We have added that expression. (Lines 228–220)

F-actin related data: authors are providing movies, Quantification included as figure in the paper would help the reading and the understanding of the actin-related phenotype? :line 237 authors mentioned F-actin dynamics, which parameters have been analysed?

Thank you for pointing this out. We included this data in Fig.4.

We apologize for the unclear wording. We meant “the dynamics of F-actin” as the localization of F-actin in the cell, so we changed the term to “F-actin localization”. (Line231)

4I: why not doing the TGF- β RII staining in *Lpar1-eGFP* to ensure the localization at SP ?

TGF β RII staining, as shown in Fig. 4I, was performed using the E15.5 stage section. The *Lpar1-EGFP* signal appears from E17 onward and is barely detectable at this stage. The location of the SP layer was therefore determined and marked from the DAPI staining pattern.

Referee #3:

I didn't ask for extensive revisions in my previous review but requested two critical data to demonstrate causality in the author's system. The epistasis tests I suggested were required to fully elucidate the role of TGF-Beta signaling in relation to ADAMTS2. The authors provided what happened when TGF-Beta gain of function

was made in the ADAMTS2 loss of function context. Curiously, the authors are conspicuously silent on the second half of the epistasis test, namely, if TGF loss of function should rescue the migration block due to ADAMTS2 gain of function. I would still require this experiment to buy into the author's proposed model fully.

Thank you for pointing this out. In the last revision, we showed that the knockdown of TGF β 2 overexpression could rescue the loss of function phenotype of Adamts2. However, we certainly did not perform the reverse rescue experiment. Therefore, in this revision, we conducted a verification experiment to determine whether the migration defect of Adamts2 overexpression can be rescued by the knockdown of TGF β signaling. We performed both overexpression of Adamts2 and shRNA knockdown of TGF- β RII under the NeuroD1 promoter. We constructed a plasmid in which sh-TGF β RII is expressed in a NeuroD1-Cre-dependent manner. As a result, as shown in Fig.S11, there was a trend toward rescue. However, there was no significant difference between the two groups, indicating that the precise regulation of transient activation of TGF β signaling by Adamts2 is critical. Although the difference was not significant, this analysis showed an increase in TGF β signaling due to the gain of function of Adamts2, suggesting that Adamts2 and TGF β signaling have an epistatic relationship.

Cross-comments by referee 1:

Just to be clear, I was satisfied with the authors' responses regarding my personal remarks which were more focused on the ADAMTS enzymes.

I am not a specialist in neurology, but it seems that the remarks of the other two reviewers are relevant and worth addressing.

Cross-comments by referee 2:

I saw that authors are willing to answer my concerns. I think this will definitively strengthen their conclusion. I would also encourage them to improve the overall quality of the figures and images provided.

Concerning the relationship between TGF- β and ADAMTS2, I also asked for the rescue of the ADAMTS2 KD phenotype by increase of TGF- β signalling in the first round of revision. Although the reverse experiment could be interesting, I am not sure it would bring much to the conclusion. However, you must note that Conny Cepko has developed strategy to express miRNA under any kind of polII promoter (PCAGGS miR30 plasmids can be found in addgene). Authors will have to replace the pCAGGS promoter by the NeuroD and insert the miRNAs; it would require maximum a month or so of cloning and then 2-3 month to complete the in utero electroporation experiments.

In this revision, we constructed a knockdown plasmid using the Cre-loxP system to knockdown TGF β R2 in a neuron-specific manner using NeuroD1 promoter and found a tendency to rescue migration defects caused by Adamts2 overexpression (Fig.S11).

Cross-comments by referee 3:

The NeuroD promoter experiment seems quite feasible, and it is an important part of the author's argument. In terms of my rationale for requesting the two experiments in my original review and requesting the TGF β loss of function in the second review: given the correlative nature of a good number of the original experiments, I asked for these epistasis experiments to get at causality in their system. I re-asked for the TGF β loss of function experiment because the authors didn't even acknowledge that specific experiment negatively or positively in their rebuttal. I potentially understand their predicament: I've had a similar situation in a recent paper submission where silencing a gene on a single-cell basis did not yield a phenotype but knocking the same gene out in a whole cell population did (a non-cell autonomous function). If they can't do my experiment without an extensive delay, I'd like to see two things 1) they should change the language of their discussion to reflect the lack of full causality in their working model, and 2) if they did the TGF β silencing but saw a negative result they should report it perhaps in the form of a supplemental figure with appropriate discussion as it relates to their working model.

As noted above, the data in this revision suggested an epistatic relationship between Adamts2 and TGF β signaling in neurons (Fig.S11). The lack of significant differences was rather expected, indicating that the precise timing of the transient increase in TGF β signaling by Adamts2 is critical in vivo and challenging to reproduce without fine control of the expression levels.

Dear Dr. Ohtaka-Marurama,

Thank you for the submission of your revised manuscript. We have now received the enclosed reports from the referees that were asked to assess it, and I am happy to say that both support its publication now. Referee 2 still has a few more minor suggestions that I would like you to address and incorporate before we can proceed with the official acceptance of your manuscript. Please submit a point-by-point response to all last comments.

A few editorial requests will also need to be addressed:

- Please add up to 5 keywords to your ms file.
- Please add a direct link to the deposited data to the Data Availability Section. Please also delete this sentence from the Acknowledgments section "The microarray data are available in Gene Expression Omnibus database under accession number GSE102911."
- Please add a "Disclosure and Competing Interest Statement" to the ms file.
- Please correct the reference format to the EMBO reports (Harvard) style. The EMBO reports style is also in EndNote.
- All main and all EV figures need to be uploaded as individual figure files.
- The main figures must fit onto one page, Fig 7 still runs over 2 pages, please correct. You could also split the figure into 2 figures, if this makes it easier.
- Each movie should be zipped together with its legend and uploaded as one individual file per movie.
- The manuscript sections are in the wrong order. The order of the ms sections should be: abstract, introduction, results, discussion, methods, data availability section, acknowledgments, disclosure statement and competing interests, references, main figure legends, EV figure legends. Please correct.
- There are 11 supplementary figures. Some run over 2 pages so are not a suitable format for an EV (expanded view) figure. You can choose 5 supplementary figures that can be uploaded as EV figures. EV figures are integrated into the ms text online and expand when clicked. The remaining figures can be combined in the Appendix file. Please see our guide to authors for more information:
<https://www.embopress.org/page/journal/14693178/authorguide#manuscriptpreparation>
Please also correct all figure callouts in the ms text.
- I attach again to this email the related ms file with comments on the figure legends added by our data editors. All comments need to be addressed.
- Table 1 needs to be called "Dataset EV1", please also correct the callout in the ms text.
- There is an author name discrepancy: Chiaki Ohtaka-Maruyama in the ms file versus Chiaki Ohtaka-Marurama in our online submission system, please correct.

I would like to suggest some changes to the title and abstract. Please let me know whether you agree with the following:

ADAMTS2 promotes radial migration by activating TGF- β signaling in the developing neocortex

The mammalian neocortex is formed by sequential radial migration of newborn excitatory neurons. Migrating neurons undergo a multipolar-to-bipolar transition at the subplate (SP) layer, where extracellular matrix (ECM) components are abundantly expressed. Here, we investigate the role of the ECM at the SP layer. We show that TGF- β signaling-related ECM proteins, and their downstream effector, p-smad2/3, are selectively expressed in the SP layer. We also find that migrating neurons express a disintegrin and metalloproteinase with thrombospondin motif 2 (ADAMTS2), an ECM metalloproteinase, just below the SP layer. Knockdown and knockout of Adamts2 suppresses the multipolar-to-bipolar transition of migrating neurons and disturbs radial migration. Time-lapse luminescence imaging of TGF- β signaling indicates that ADAMTS2 activates this signaling pathway in migrating neurons during the multipolar to bipolar transition at the SP layer. Overexpression of TGF- β 2 in migrating neurons partially rescues migration defects in ADAMTS2 knockout mice. Our data suggest that ADAMTS2 secreted by the migrating multipolar neurons activates TGF- β signaling by ECM remodeling of the SP layer, which might drive the multipolar to bipolar transition.

EMBO press papers are accompanied online by A) a short (1-2 sentences) summary of the findings and their significance, B) 2-3 bullet points highlighting key results and C) a synopsis image that is exactly 550 pixels wide and 200-600 pixels high (the height is variable). Please show a model, like a graphical abstract, in the synopsis image. Please note that all text needs to be readable at the final image size. Please send us this information along with the final manuscript.

Referee #2:

I acknowledge the effort of the authors to address my concerns.

I am now more convinced about the cell autonomous effect and the reliability of the counting.

For the rescue experiment now presented in fig S9, quantification show that ADAMTS3 fully rescues the migration phenotype while ADAMTS14 does not. Accordingly, the authors should change the sentence line 225 by removing "possibly Adams14" as their results do not show that.

I suggested to assess the effect of KD and overexpression of ADAMTS2 using a neuroD promotor. Authors have performed the overexpression experiment that are very convincing. They argue that cell specific expression of siRNA is not possible. Although this is correct, they could have used the same strategy used in Figure S11, meaning the CRE inducible expression of shRNA. Would have been an added value.

Clarification required in the text:

Figure 1C/ line 117: could the authors clarify when the time lapse starts after the ChaseABC treatment. I don't think it is after the 3 days of treatment.

S6C-E: what mean the different colours used to show the segmented cells (orange and yellow dashed lines)?

Figure 4: what is the green staining - not said neither on the figure nor in the legend (lifeact??)

Figure 1E/S2: as the authors have the data, would have been much more interesting to present a time series of migrating neurons instead of a single time point: it would have addressed the dynamic of gelatinolytic activities in migrating neurons.

Referee #3:

The authors have attempted to address my remaining concerns. The new inclusions provide stronger support for their model and is now suitable for publication.

Response to Reviewer#2

We thank the referee for his/her valuable comments, which helped to improve our manuscripts. We carefully considered all comments.

Referee #2:

I acknowledge the effort of the authors to address my concerns.

I am now more convinced about the cell autonomous effect and the reliability of the counting.

Thank you.

For the rescue experiment now presented in fig S9, quantification show that ADAMTS3 fully rescues the migration phenotype while ADAMTS14 does not. Accordingly, the authors should change the sentence line 225 by removing "possibly Adamts14" as their results do not show that.

Thank you for pointing this out. We have changed that sentence as you mentioned.

Before:

「These results suggest that Adamts3 and possibly Adamts14 are involved in the compensation mechanism in the homozygous Adamts2 KO mice.」

After:

「These results suggest that Adamts3 is involved in the compensation mechanism in the homozygous Adamts2 KO mice.」 (Line230)

I suggested to assess the effect of KD and overexpression of ADAMTS2 using a neuroD promotor. Authors have performed the overexpression experiment that are very convincing. They argue that cell specific expression of siRNA is not possible. Although this is correct, they could have used the same strategy used in Figure S11, meaning the CRE inducible expression of shRNA. Would have been an added value.

We initially tried several sh-RNA constructs, but the knockdown efficiency was higher with siRNA when we verified with cultured cells. Therefore, we conducted the KD experiments using siRNA. Then we studied the conditions for knockdown, examined the optimal concentration of KD efficiency, and performed rescue experiments by

titrating with Adamts2 resistant to siRNA in different ratios to overexpression. As you say, it is possible to conduct experiments with NeuroD1-Cre to express sh-RNA constructs, and we may find suitable sh-RNA constructs for it in the future. We want to try it out in the future.

Clarification required in the text:

Figure 1C/ line 117: could the authors clarify when the time lapse starts after the CHaseABC treatment. I don't think it is after the 3 days of treatment.

We are sorry for not being clear on the time description. We performed 50 hours of imaging after adding CHaseABC, as shown in the attached reference figure, of which 3 hours of images from around 32.5-hour time points are shown in Fig. 1C.

We have described it in the figure legend of Fig 1C as follows.

「Time-lapse imaging was performed for 50 h after adding CHaseABC; images for three h after 32.5h are shown」(Line794-796)

S6C-E: what mean the different colours used to show the segmented cells (orange and yellow dashed lines)?

I apologize for the missing explanatory note for these. The following explanation has been added to figure legend of Appendix figureS4.

「Cells circled by white dashed lines are positive for GFP only, yellow lines are GFP positive and double positive for Ki67 or Tbr2, and orange dashed lines are cells positive for Ki67 or Tbr2 alone.」

Figure 4: what is the green staining – not said neither on the figure nor in the legend (lifeact??)

Sorry for the confusion. Green is Lifeact. We have added that to the legend.

Figure 1E/S2: as the authors have the data, would have been much more interesting to present a time series of migrating neurons instead of a single time point: it would have addressed the dynamic of gelatinolytic activities in migrating neurons.

Thanks for the comment. We electroporated RFP at E14, made slices with E17, and

did time-lapse imaging for 20 hours with DQ-gelatin in the medium. We found that only the migrating cells passing through the SP layer showed specific gelatinase activity and that the red migrating cells had green indicator signals of gelatinase activity. Therefore, a picture of another imaging was added to EV Fig.2. In EV Fig .2-B, if you zoom in on area a, cells before SP entry (cell 1) are green negative, but cells in SP (cells 2, 3 and 4) become green positive and show gelatinase activity. However, cells that have also passed through the SP layer (cells 5,6) are green-negative. Thus, as you can see from this image, migrating cells specifically show increased gelatinase activity in the SP layer.

Dr. Chiaki Ohtaka-Maruyama
Tokyo Metropolitan Institute of Medical Science
Department of Brain Development and Neural Regeneration
2-1-6 Kamikitazawa
Setagaya-ku
Tokyo 156-8506
Japan

Dear Dr. Ohtaka-Maruyama,

I am very pleased to accept your manuscript for publication in the next available issue of EMBO reports. Thank you for your contribution to our journal.

I slightly modified your short summary and bullet points. Do you agree with this:

ADAMTS2 is secreted by migratory neurons during the formation of the mammalian neocortex. ADAMTS2 promotes neuronal migration by activating TGF- β signaling in migrating neurons during the multipolar-to-bipolar transition at the SP layer.

- ECM proteins are abundant in the subplate layer of the neocortex.
- Admts2 is upregulated during the migration of newborn neurons in the developing cortex.
- ADAMTS2 activates TGF- β signaling perhaps by ECM remodeling of the SP layer, which might facilitate the multipolar to bipolar transition.

Yours sincerely,

Corresponding Author Name: Chiaki Ohtaka-Maruyama
Journal Submitted to: EMBO Reports
Manuscript Number: EMBOR-2022-55992-T

USEFUL LINKS FOR COMPLETING THIS FORM

The EMBO Journal - Author Guidelines
EMBO Reports - Author Guidelines
Molecular Systems Biology - Author Guidelines
EMBO Molecular Medicine - Author Guidelines